# Temporal derivative computation in the dorsal raphe network revealed by an experimentally driven augmented integrate-and-fire modeling framework

Emerson F Harkin[1], Michael B Lynn[1], Alexandre Payeur[1,2†], Jean-François Boucher[1], Léa Caya-Bissonnette[1], Dominic Cyr[1], Chloe Stewart[1], André Longtin[1,2], Richard Naud[1,2]*, Jean-Claude Béïque[1]*

[1]Brain and Mind Research Institute, Centre for Neural Dynamics, Department of Cellular and Molecular Medicine, University of Ottawa, Ottawa, Canada; [2]Department of Physics, University of Ottawa, Ottawa, Canada

**\*For correspondence:**
rnaud@uottawa.ca (RN);
jbeique@uottawa.ca (J-CB)

**Present address:** †Mila,
Université de Montréal,
Montréal, Canada

**Competing interest:** The authors declare that no competing interests exist.

**Abstract** By means of an expansive innervation, the serotonin (5-HT) neurons of the dorsal raphe nucleus (DRN) are positioned to enact coordinated modulation of circuits distributed across the entire brain in order to adaptively regulate behavior. Yet the network computations that emerge from the excitability and connectivity features of the DRN are still poorly understood. To gain insight into these computations, we began by carrying out a detailed electrophysiological characterization of genetically identified mouse 5-HT and somatostatin (SOM) neurons. We next developed a single-neuron modeling framework that combines the realism of Hodgkin-Huxley models with the simplicity and predictive power of generalized integrate-and-fire models. We found that feedforward inhibition of 5-HT neurons by heterogeneous SOM neurons implemented divisive inhibition, while endocannabinoid-mediated modulation of excitatory drive to the DRN increased the gain of 5-HT output. Our most striking finding was that the output of the DRN encodes a mixture of the intensity and temporal derivative of its input, and that the temporal derivative component dominates this mixture precisely when the input is increasing rapidly. This network computation primarily emerged from prominent adaptation mechanisms found in 5-HT neurons, including a previously undescribed dynamic threshold. By applying a bottom-up neural network modeling approach, our results suggest that the DRN is particularly apt to encode input changes over short timescales, reflecting one of the salient emerging computations that dominate its output to regulate behavior.

## Editor's evaluation

To characterize physiological properties of dorsal raphe serotonin neurons, the authors applied the approach called an augmented generalized integrate-and-fire [aGIF] model, which incorporates a relatively small number of salient biophysical properties of a specific neuron type, and whose parameters are optimized based on voltage dynamics obtained experimentally. The results showed that after-hyperpolarization and A-type potassium currents, in combination with heterogeneous feedforward inhibition from local GABA neurons, give rise to a derivative-like input-output relationship in serotonin neurons.

## Introduction

The forebrain-projecting serotonin (5-HT) neurons of the dorsal raphe nucleus (DRN) play a key role in regulating behavior in dynamic environments, but the precise nature of this role is still not well understood (*Young et al., 1985*; *Delgado, 1994*; *Warden et al., 2012*; *Dayan and Huys, 2015*). DRN serotonin neurons have been proposed to modulate a wide range of cognitive processes, such as encouraging patience for future rewards (*Miyazaki et al., 2014*; *Fonseca et al., 2015*), signaling the beneficialness of current actions or states (*Luo et al., 2016*), complementing reinforcement signals of dopamine (*Daw et al., 2002*; *Maier and Watkins, 2005*; *Nakamura et al., 2008*; *Ranade and Mainen, 2009*; *Tops et al., 2009*; *Cools et al., 2011*; *Li et al., 2016*), and, partially as a corollary, regulating both learning (*Soubrié, 1986*; *Deakin, 1991*; *Daw et al., 2002*; *Dayan and Huys, 2009*; *Matias et al., 2017*; *Grossman et al., 2022*) and mood (*Savitz et al., 2009*; *Fava and Kendler, 2000*; *Donaldson et al., 2013*; *Cipriani et al., 2018*). While the remarkable diversity of roles attributed to this single neurotransmitter has historically been perplexing, recent findings are beginning to provide insight (see *Okaty et al., 2019* for review). For example, the unsuspected organization of 5-HT neurons into anatomical sub-modules that differentially regulate behavior (*Abrams et al., 2004*; *Lowry et al., 2005*; *Commons, 2015*; *Muzerelle et al., 2016*; *Ren et al., 2018*; *Paquelet et al., 2022*), or the observation that 5-HT neurons can encode distinct salient features of the environment over different timescales (*Trulson and Jacobs, 1979*; *Schweimer and Ungless, 2010*; *Ranade and Mainen, 2009*; *Cohen et al., 2015*; *Zhong et al., 2017*) is a compelling mechanism that may contribute to the multiplicity of 5-HT's actions. These anatomical and dynamical perspectives on 5-HT diversity need not be mutually exclusive. A clearer understanding of the biophysical mechanisms that contribute to the coding features of raphe neurons over multiple timescales has the potential to substantially increase our understanding of how 5-HT regulates behavior.

The spiking statistics of 5-HT neurons necessarily shape and constrain their computational role. For instance, the slow firing rate (~5 Hz) of 5-HT neurons, in large part attributable to a large after-hyperpolarization potential (AHP) (*Aghajanian and Vandermaelen, 1982*; *Vandermaelen and Aghajanian, 1983*), may appear to preclude signaling on faster timescales. However, fast signaling despite slow firing can arise naturally in ensemble-rate codes (*Knight, 1972*; *Gerstner, 2000*). Consistent with this idea, the in vivo population activity of 5-HT neurons has been observed to track impending rewards over second to sub-second timescales (*Zhong et al., 2017*), and the trial-averaged ensemble rates of individual 5-HT neurons can track environmental changes over the millisecond timescale (*Ranade and Mainen, 2009*; *Cohen et al., 2015*). In addition, the fact that 5-HT receptor subtypes can regulate the excitability of target neurons over different timescales, including ionotropic 5-HT3 receptors with millisecond gating kinetics (*Béïque et al., 2004*; *Béïque et al., 2007*; *Andrade, 2011*; *Varga et al., 2009*), at the very least suggests that the 5-HT system is capable of fast information transmission, an observation mirrored by the fast dynamics of neurons which project to the DRN (*Amo et al., 2014*; *Matsumoto and Hikosaka, 2007*). If fast and slow signaling by the DRN are manifest, it is less clear which cellular mechanisms regulate the interplay between these timescales, nor which input features are represented on which timescales.

Computational modeling is a standard approach to link levels of description and is thus well suited to delineate how network-level function emerges from excitability features identified at the single-cell level. In spite of their conceptual utility, the most detailed single cell models, including those of DRN neurons (*Tuckwell and Penington, 2014*; *Wong-Lin et al., 2011*), do not lend themselves with ease to bottom-up modeling efforts because of the substantial technical difficulty of obtaining sufficiently accurate values for a large number of interacting model parameters (*Prinz et al., 2004*; *Gerstner and Naud, 2009*). Mathematically simpler generalized integrate-and-fire (GIF) models provide a strong foundation for network modeling because their small number of parameters can be estimated with a high degree of precision (*Mensi et al., 2012*; *Pozzorini et al., 2013*; *Teeter et al., 2018*). This precision comes at a price, however the process of distilling the effects of many biophysical mechanisms into a small number of model parameters makes it difficult to study a specific mechanism (e.g. a subthreshold ion channel) in isolation. A hybrid approach based on a reductionistic GIF model augmented with a limited set of biophysical mechanisms could leverage the precision of GIFs while allowing the ability to link specific biophysical mechanisms with higher-order network function.

In this study, we developed and validated for DRN neurons a hybrid modeling approach that lies between reductionist GIF and biophysical Hodgkin-Huxley-type models to capture excitability

features of individual neurons for accurate simulations of population dynamics and, by extension, network computation inference. To this end, we carried out cellular electrophysiological recordings from genetically identified DRN 5-HT and SOM neurons to (1) extract and validate, from sets of noisy inputs, parameters for the automatic development of accurate GIF models and (2) experimentally define complementary biophysical mechanisms to be grafted onto the GIF models to iteratively improve their prediction accuracy (*augmented* GIFs). This approach recapitulated and extended past findings on DRN neurons by showing that the best-performing models of 5-HT neurons featured slow membrane time constants, an A-type potassium current, and strong adaptation mechanisms. Network simulations of optimized GIF models of both 5-HT and GABAergic SOM neurons organized in a feed-forward inhibitory circuit revealed that 5-HT neuron populations context-dependently encode a mixture of the intensity and temporal derivative of their inputs. Our overall approach further allowed us to trace back specific features of these population responses (e.g. gain) to defined excitability features of DRN neurons.

## Results

### Salient electrophysiological features of DRN neurons

Our main goal was to develop an experimentally grounded model of the DRN to better understand its computational properties. As a first step toward this goal, we carried out experiments to constrain a set of single-neuron models of the two main cell types found in the DRN: 5-HT and SOM GABA neurons. We performed whole-cell electrophysiological recordings from genetically identified 5-HT (*Figure 1A1*; SERT-Cre::Rosa-TdTomato mice) and SOM (*Figure 1A2*; *Table 1*; SOM-Cre::Rosa-TdTomato mice) neurons in slices. In keeping with previous descriptions (e.g. *Vandermaelen and Aghajanian, 1983*; *Calizo et al., 2011*), in the majority of the 5-HT neurons recorded in our dataset, current steps induced strongly adapting action potential firing accompanied by large AHPs, and a characteristic kink in the voltage trace leading up to the first spike (*Figure 1B*). Qualitatively distinct firing patterns of 5-HT neurons were, however, occasionally observed (*Figure 1—figure supplement 1*). Recordings from SOM neurons revealed spiking patterns that were more heterogeneous (*Figure 1—figure supplement 2*). Comparing the relationship between the injected currents and firing frequencies between these populations, we found that SOM neurons were generally more sensitive to changes in input current (gain) than 5-HT neurons and responded to weaker inputs (*Figure 1B*). The gain showed greater variability in SOM neurons than in 5-HT neurons (Brown-Forsythe equality of variance test p=0.001 on N = 17 5-HT and N = 7 SOM neurons). In line with this observation, SOM neurons also consistently exhibited a wider range of firing frequencies for a given input (e.g. for a 50 pA input 5-HT neurons fired at 2.81 ± 2.22 Hz vs 8.16 ± 5.70 Hz; Brown-Forsythe test p=0.005 in $N = 17$ 5-HT neurons and $N = 14$ SOM cells). Together, these observations outlined three salient cellular-level features of DRN neurons, namely the strong AHP and voltage kink of 5-HT neurons as well as noticeable heterogeneous excitability of SOM neurons.

The characteristic kink in the voltage leading up to the first spike in 5-HT neurons in principle may be caused by near-threshold activation of voltage-gated potassium channels (VGKCs; *Connor and Stevens, 1971*; *Connor et al., 1977*; *Drion et al., 2015*). We therefore examined whole-cell currents evoked by voltage steps (from –90 mV to –20 mV) in both 5-HT and SOM neurons to look for evidence of such a VGKC. In 5-HT cells, these experiments revealed a large (peak amplitude 928 ± 249 pA, leak-subtracted), partly inactivating (steady-state amplitude 142 ± 45 pA, leak-subtracted) outward current (*Figure 1C1*) that was sensitive to K$_v$4-selective potassium channel blockers (*Figure 1—figure supplement 4*). This current activated rapidly (peak latency 7.46 ± 0.21 ms) and inactivated over tens of milliseconds (inactivation time constant $\tau_h = 42.9 \pm 9.4$ ms; kinetics are similar at near-physiological temperature, see *Figure 1—figure supplement 5*). The gating and kinetic profile (*Table 2*, *Figure 1—figure supplements 6 and 7*) of the inactivating component of this conductance in 5-HT neurons are broadly similar to those expected of the A-type potassium currents ($I_A$) characterized in great detail in several other cell types (e.g. *Storm, 1989*). Because these parameters are sufficient to construct a model of this conductance (see below), we have not attempted to determine its molecular identity further. For the sake of simplicity, we refer to the inactivating component herein as $I_A$ (in keeping with the previous literature; see *Aghajanian, 1985*; *Tuckwell and Penington, 2014*) and the steady-state component as $I_K$. Thus, an $I_A$-like inactivating VGKC is a consistent feature of DRN 5-HT neurons.

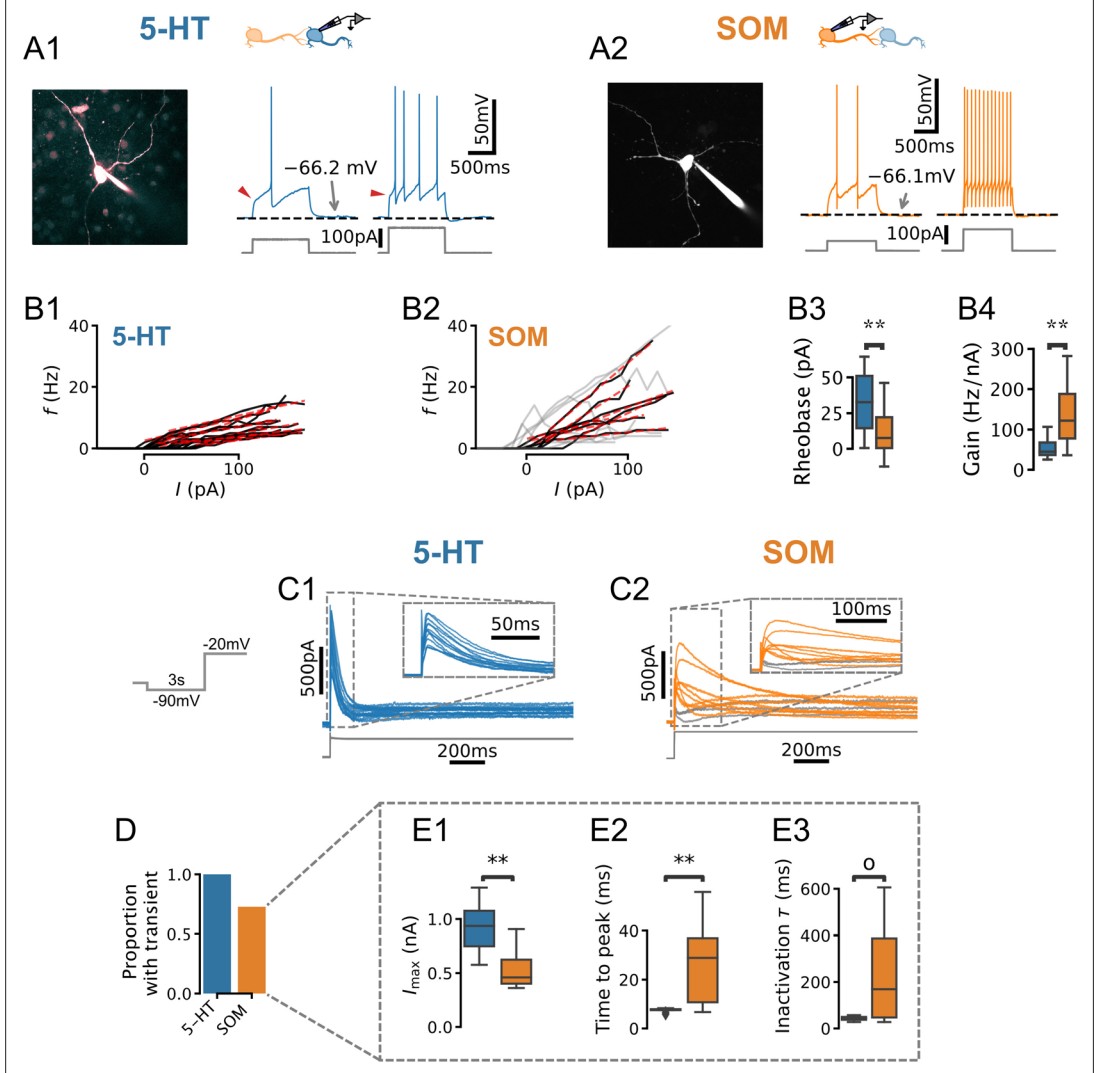

**Figure 1.** Physiology of dorsal raphe nucleus (DRN) neurons. (**A, B**) Morphology, current steps (**A**), and spike frequency vs. input ($f/I$) curves (**B**) of genetically identified DRN neurons. Non-monotonically increasing $f/I$ curves (gray) and linear fits to monotonically increasing curves (red dashed) for $N = 17$ serotonin (5-HT) (**B1**) and $N = 14$ somatostatin (SOM) (**B2**) neurons. (**B3**) Rheobase of 33.8 ± 21.0 pA in 5-HT neurons vs. 11.3 ± 16.0 pA in SOM neurons. (**B4**) Gain of 52.2 ± 22.2 Hz/nA in 5-HT neurons vs. 87.2 ± 33.0 Hz/nA in $N = 7$ SOM neurons with monotonically increasing $f/I$ curves. (**C**) Leak-subtracted whole-cell currents evoked by a depolarizing step. Each trace is one cell; $N = 13$ 5-HT and $N = 11$ SOM cells. Traces without a transient outward current are shown in gray. (**D**) Proportion of neurons with a transient outward current by cell type. (**E**) Quantification of transient outward currents in each cell type. $N = 3$ SOM cells without a transient outward current were excluded from analysis, leaving $N = 13$ 5-HT and $N = 8$ SOM neurons. Annotations reflect Mann-Whitney U-tests. Non-parametric Brown-Forsythe equality of variance tests indicated significantly more variable time to peak (p=1.11e-4; **E2**) and inactivation time constant (p=1.97e-4; **E3**) in SOM cells.

The online version of this article includes the following figure supplement(s) for figure 1:

**Figure supplement 1.** Firing patterns of four positively identified serotonin (5-HT) neurons (**A-D**).

**Figure supplement 2.** Dorsal raphe nucleus (DRN) somatostatin (SOM) neurons are not homogenous.

**Figure supplement 3.** Full distributions of membrane parameters listed in *Table 1*.

**Figure supplement 4.** Transient outward current found in serotonin (5-HT) cells ($I_A$) is sensitive to potassium channel blockers.

**Figure supplement 5.** Temperature-dependence of $I_A$ amplitude and kinetics in serotonin (5-HT) neurons.

**Figure supplement 6.** Characterization of voltage-dependence of $I_A$ in serotonin (5-HT) neurons.

**Figure supplement 7.** Temperature-dependence of the gating of $I_A$.

**Table 1.** Membrane parameters of DRN neurons.

Parameters obtained from recordings from mPFC L5 pyramidal neurons used to fit GIF models as a point of comparison are also shown. Data are presented as mean ± SD. Distributions are shown in *Figure 1—figure supplement 3*.

| Cell type | R (GΩ) | C (pF) | τ (ms) | N |
|---|---|---|---|---|
| 5-HT | 1.16±0.55 | 67.0±17.1 | 75.2±33.8 | 96 |
| SOM | 1.07±0.58 | 43.5±15.5 | 42.2±19.8 | 28 |
| mPFC | 0.188±0.130 | 160.6±48.2 | 27.4±13.2 | 25 |

DRN, dorsal raphe nucleus; PFC, prefrontal cortex; GIF, generalized integrate-and-fire model; 5-HT, serotonin; SOM, somatostatin; mPFC, medial prefrontal cortex; R, memrane resistance; C, membrane capacitance; $\tau$, membrane time constant.

The same voltage-clamp protocol applied to SOM neurons, in contrast, triggered a mixture of outward and inward currents that varied widely from cell to cell (*Figure 1D2*). A significant proportion of SOM neurons did not express a transient outward current at all (27.3 %, *Figure 1E*), while the remaining cells had currents that were significantly smaller ($p = 0.003$), activated more slowly ($p = 0.003$), and exhibited much more heterogeneous kinetic profiles than those found in 5-HT neurons (*Figure 1E3*). Together, these results show that the expression of this subthreshold voltage-gated current is substantially more variable in SOM neurons than in 5-HT neurons, in line with the distinctive heterogeneity of excitability features observed in this DRN cell type (*Figure 1C2*, *Figure 1—figure supplement 2*).

## $I_A$ regulates initial firing rate via a control of spike time jitter

To develop an intuition for how $I_A$ impacts the firing patterns of 5-HT populations, we first created a toy leaky integrate-and-fire (LIF) model that captured the effect of this conductance on single-cell voltage dynamics (see Methods). In keeping with previous studies, $I_A$ introduced a kink in the subthreshold voltage leading up to spike threshold (*Getting, 1983*; *Segal, 1985*; *McCormick, 1991*; *Figure 2A*) and increased the latency to the first spike evoked by a square step stimulus, particularly when starting from a hyperpolarized voltage at which $I_A$ is free from inactivation (*Figure 2B and C*). The effect of $I_A$ on spike latency depends at least to some extent on its effective magnitude and inactivation kinetics (defined as the ratio of maximal A-type conductance to inverse membrane resistance and the ratio of the inactivation time constant of $I_A$ to the membrane time constant; see Methods). When we set the corresponding parameters in our toy model to experimentally determined values from 5-HT neurons, we observed the same qualitative relationship between spike latency and initial voltage (*Figure 2—figure supplement 1*), further pointing toward a functional effect of $I_A$ in this cell type. The predicted relationship between initial voltage and latency was experimentally recapitulated in whole-cell recordings from identified 5-HT neurons (*Figure 2D–F*). In particular, the onset of spiking was delayed by hyperpolarization (*Figure 2D*), and the magnitude of this effect was significantly

**Table 2.** 5-HT $I_A$ current gating parameters.

Gating curves shown in *Figure 1—figure supplement 6B* were fitted with the scaled Boltzmann function $g_\infty(V)/g_\infty(V_{ref}) = x_\infty = A_x/\left(1 + \exp[-k_x(V - V_x^*)]\right)$. Values are based on experiments from N=13 cells.

| Gate | $V_{ref}$ (mv) | $A_x$ | $k_x$ (mV$^{-1}$) | $V_x^*$ (mV) |
|---|---|---|---|---|
| $m_\infty$ | −20 | 1.61 | 0.0985 | −23.7 |
| $h_\infty$ | −80 | 1.03 | −0.165 | −59.2 |
| $n_\infty$ | −20 | 1.55 | 0.216 | −24.3 |

5-HT, serotonin.

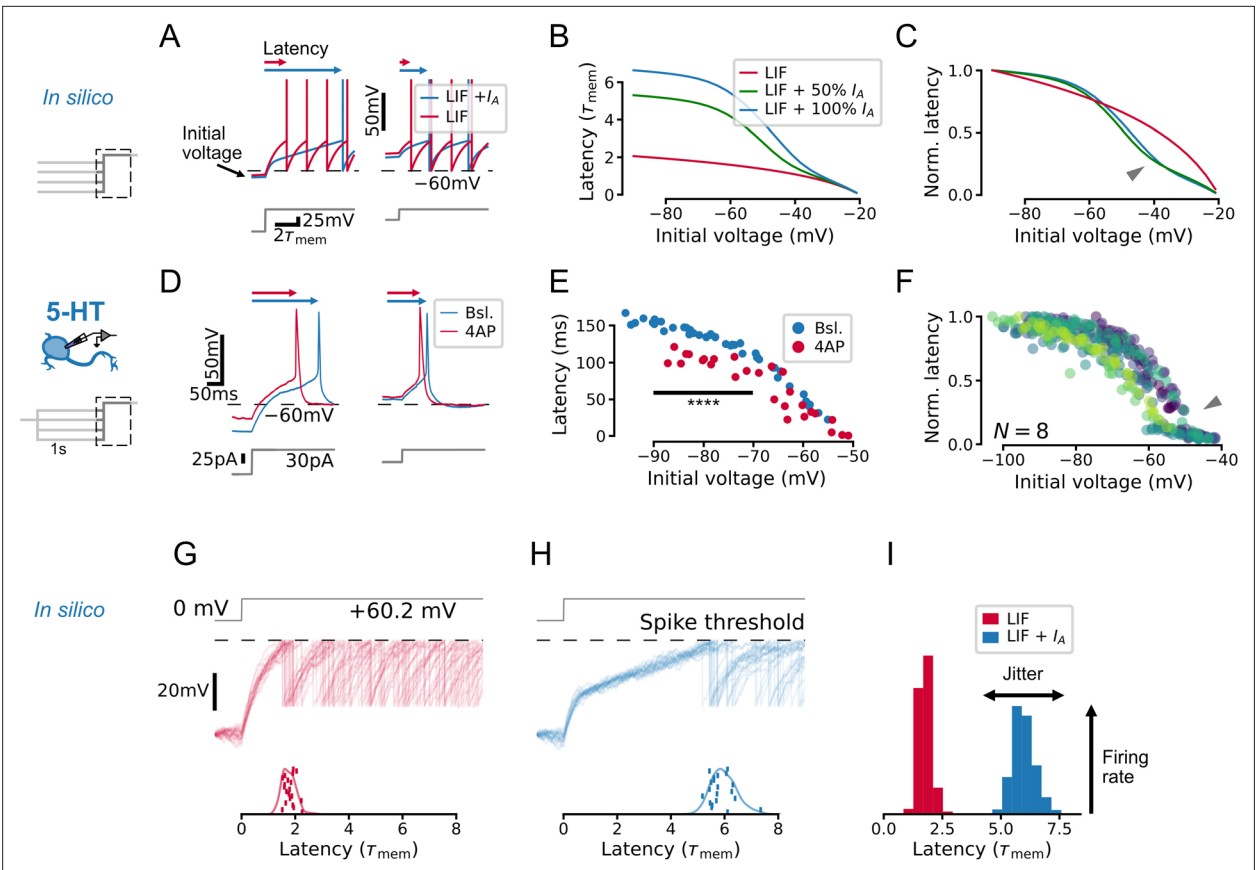

**Figure 2.** $I_A$ qualitatively alters the relationship between initial voltage and spike timing. (**A–C**) A toy leaky integrate-and-fire (LIF) model neuron with $I_A$ predicts a non-linear effect of voltage history on spike timing in a simple experiment. (**D–F**) Experiments in serotonin (5-HT) neurons fulfill predictions of the toy model. (**F**) Latency curves for $N = 8$ 5-HT neurons, normalized to the maximum latency for each cell. Each color is one cell. (**G–I**) $I_A$ causes an increase in spike latency and jitter in the presence of noise. Models and input are the same as in A–C. Spike latency histograms for populations of 600 toy neurons are shown in I. The width of the histogram reflects jitter in the timing of the first spike, while the height of the histogram approximates the peak instantaneous firing rate. Note that as jitter increases, the height of the histogram decreases. The toy model with 100% $I_A$ has an effective $I_A$ conductance $\bar{g}'_A/g_l = 10$ and effective inactivation time constant $\tau_h/\tau_{mem} = 1.2$.

The online version of this article includes the following figure supplement(s) for figure 2:

**Figure supplement 1.** Parameter and temperature-dependence of the effect of $I_A$ on spike-timing.

reduced by the partial pharmacological block of $I_A$ with 4-AP ( p=3.2e-6 for initial voltages between –90 mV and –70 mV; *Figure 2E*). Finally, we also observed an inflection point predicted by the model in the normalized initial-voltage/latency relationship (*Figure 2F*, compare with model prediction in *Figure 2C*). In summary, our toy model captured the expected effects of $I_A$ in single 5-HT cells.

Next, we used our experimentally validated toy model to understand how $I_A$ impacts the spiking responses of whole neuronal populations. To do this, we simulated the effect of a shared step input to a population of 600 toy neurons each receiving independent background noise (corresponding to naturalistic fluctuations in synaptic inputs). Whereas subthreshold fluctuations yielded time-locked spikes without $I_A$ (*Figure 2G*), they induced spiking with larger jitter across the simulated population when $I_A$ was present (*Figure 2H*). This desynchronizing effect of $I_A$ also decreased the peak population rate at the time corresponding to the mean latency (*Figure 2I*) since the peak rate corresponds to the coincidence rate from an ensemble of cells with similar properties. (The same effects were also observed in the toy models with parameters constrained to experimentally determined values; *Figure 2—figure supplement 1*.) Taken together, results from these toy models revealed a role of $I_A$ in regulating the degree of synchronization of a population following sudden inputs, suggesting that $I_A$ may regulate the gain of the DRN network to time-varying inputs. This intuition gleaned from this toy model is examined in more detail with optimized GIF models (see below).

## Extensions to GIF models are required to capture the excitability of DRN neurons

We next sought to develop a model able to capture the essential biophysical features of DRN neurons and accurately predict their responses to naturalistic inputs. GIF models offer a flexible modeling framework well suited to this purpose because they can be trained to accurately reproduce the firing patterns of individual neurons using less than 5 min of electrophysiological data per neuron (*Gerstner et al., 2014*; *Teeter et al., 2018*; *Paninski et al., 2005*; *Mensi et al., 2012*; *Pozzorini et al., 2015*). In this framework, individual neurons are described in terms of three core components: (1) a passive membrane filter, $\kappa$, which transforms input currents into a subthreshold membrane potential; (2) a stochastic spiking process, which transforms the subthreshold membrane potential into action potentials; and (3) two adaptation mechanisms, namely a spike-triggered current mediating the commonly observed AHP, $\eta$, and change in firing threshold, $\gamma$ (*Figure 3A1*, see Methods). These components are described by parameters, the values of which are inferred from the electrophysiological data using a combination of least-squares multi-linear regression and gradient ascent of a likelihood function. The flexibility and data efficiency of this framework lend itself well to capturing the functional properties of single neurons and, by extension, heterogeneous neural populations.

Our results outlined in *Figure 2* show that $I_A$ regulates spike timing in 5-HT neurons because of its nonlinear subthreshold effects. Foreseeing that the presence of this prominent current may limit the accuracy of canonical GIF models—which are not designed to capture nonlinear subthreshold effects—we first *augmented* the canonical GIF model (aGIF; *Figure 3A2*) with a simplified Hodgkin-Huxley-type model of the subthreshold voltage-dependent currents we recorded in 5-HT neurons (see Methods). To assess whether incorporating additional biophysical details into the aGIF model might further improve its predictive performance, we turned to the previously described sodium channel-inactivation GIF model (iGIF; *Figure 3A3*), which extends the GIF model of *Mensi et al., 2012* by adding a non-parametric voltage coupling function to the dynamic spike threshold (*Mensi et al., 2016*; see Methods). Although this GIF model extension was initially conceived specifically to capture the influence of subthreshold sodium channel inactivation on firing threshold (hence, its name), the non-parametric definition of the threshold coupling function gives it the capacity to account for a wide range of other subthreshold biophysical mechanisms which regulate spiking, notably including, but not limited to, $I_A$. Comparing the performance of the more parsimonious aGIF model to that of the iGIF model enabled us to assess whether accounting for additional mechanisms that regulate spiking beyond $I_A$ might further improve our DRN neuron models.

To establish comparative GIF model benchmarks across cell types, we carried out whole-cell electrophysiological recordings not only from DRN 5-HT and SOM cells but also from canonical deep-layer pyramidal neurons of the medial prefrontal cortex (mPFC). For each recording, we applied two distinct instantiations of noisy in vivo-like inputs (see Supplemental methods, *Figure 3—figure supplement 1*), one of which was used to determine the model parameters while the other was reserved for post hoc evaluation of the models' accuracy (i.e. 'training' data and 'validation' data, respectively; see *Figure 3C*). Accuracy was assessed by comparing models with recorded data across cell types in terms of *Figure 3D*: (1) subthreshold voltage changes on training data, $R^2$, and; (2) spike timing on validation data, $M_d^*$ (where $M_d^* = 1$ is the best possible performance and $M_d^* = 0$ is the chance level; see Methods).

The canonical GIF model predicted both the subthreshold dynamics and spike timing of mPFC pyramidal neurons with high accuracy ($R^2 = 0.431 \pm 0.249$; $M_d^* = 0.783 \pm 0.134$; *Figure 3E*), consistent with previous reports on cortical pyramidal neurons (*Mensi et al., 2012*; *Pozzorini et al., 2015*; *Mensi et al., 2016*; *Teeter et al., 2018*). While our aGIF model slightly better predicted the voltage of mPFC neurons ($R^2 = 0.544 \pm 0.280$, $p = 0.028$, *Figure 3E1*), this did not translate into more accurate spike predictions ($M_d^* = 0.743 \pm 0.180$, $p = 0.710$, *Figure 3E2*), consistent with the observation that $I_A$ is not a significant conductance recorded from the cell body of mPFC pyramidal neurons (*Figure 3—figure supplement 2* and see *Dong and White, 2003*; *Dong et al., 2005*). On the basis of spike timing prediction, the canonical GIF model thus offered the most parsimonious account of the behavior of mPFC neurons.

With this point of comparison established, we next quantified the performance of each of our candidate GIF models (GIF, aGIF, and iGIF) in 5-HT neurons. As previously intuited, the canonical GIF model performed rather poorly in 5-HT neurons (*Figure 3F*), predicting <15% of the variance of the

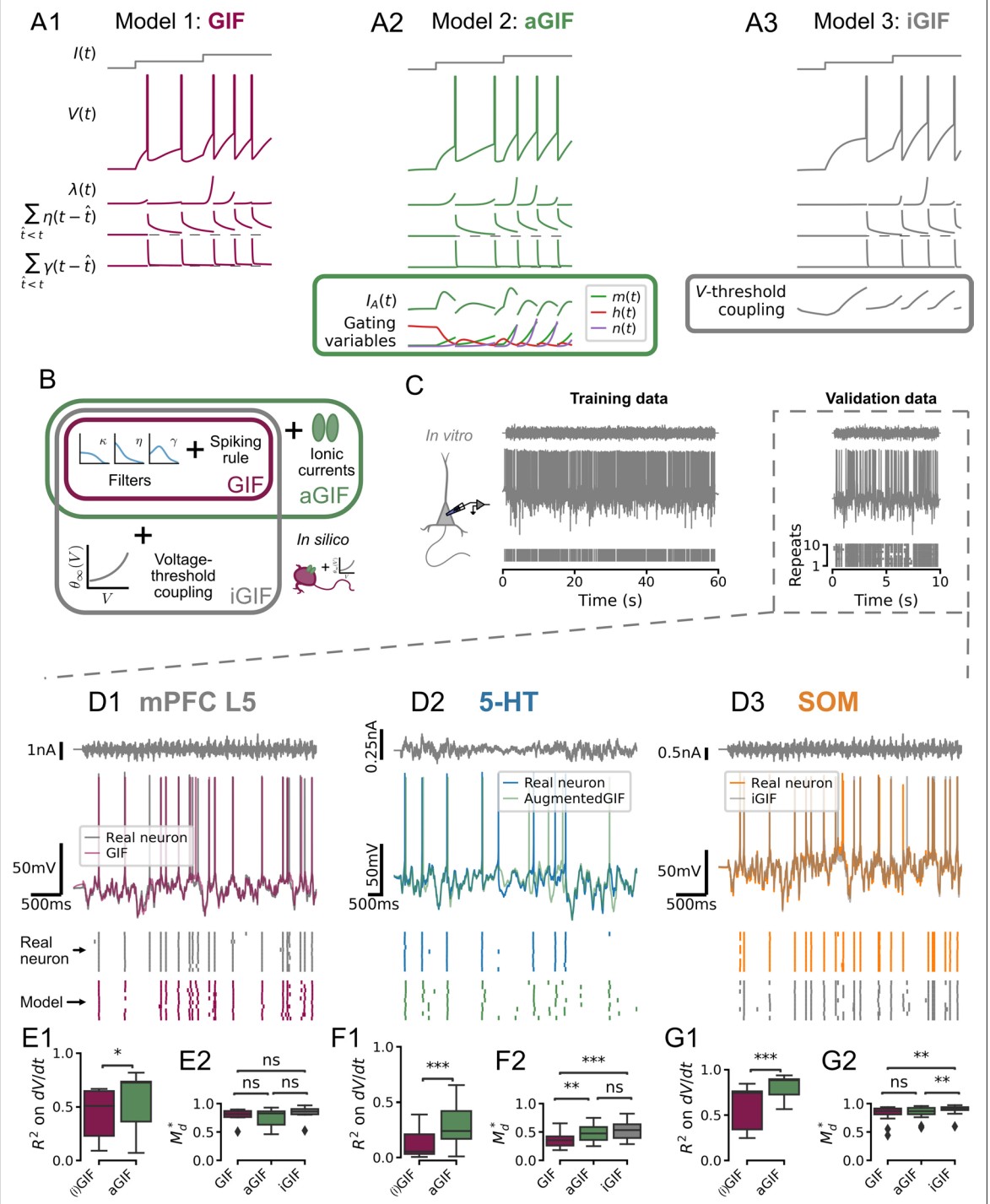

**Figure 3.** Single neuron models accurately predict the subthreshold voltage and spike times of dorsal raphe nucleus neurons. (**A**) Components of candidate single neuron models: $\lambda$ intensity of stochastic spike-generating process; $\eta$ spike-triggered current (positive values indicate a hyperpolarizing current); $\gamma$ spike-triggered threshold movement. (**B**) Generalized integrate-and-fire (GIF) model extensions. (**C**) Representative experiment used to train and validate neuron models. Training set consists of repetitions of 60 s of frozen Ornstein-Uhlenbeck (OU) noise, and the validation set consists of repetitions of a different frozen OU noise stimulus lasting 10 s (only one repetition is shown). (**D**) Representative validation data and model predictions for each cell type. OU noise input current (top), recorded and predicted voltage traces (middle), and recorded and predicted spike times across all repetitions of the validation stimulus. Stimulus parameters were adjusted for each cell type, see *Figure 3—figure supplement 1*. (**E–G**) Quantification of model performance in terms of $R^2$ on the training subthreshold voltage derivative and on the validation spike-train similarity metric $M_d^*$. GIF and inactivation GIF (iGIF) models have the same subthreshold performance because the subthreshold components of these models are identical (see

*Figure 3 continued on next page*

*Figure 3 continued*

Methods). Benchmarks are for models fitted to $N = 18$ serotonin (5-HT), $N = 14$ somatostatin (SOM), and $N = 7$ medial prefrontal cortex (mPFC) neurons. aGIF, augmented GIF.

The online version of this article includes the following figure supplement(s) for figure 3:

**Figure supplement 1.** Representative training and validation sets for all cell types.

**Figure supplement 2.** Whole-cell currents observed in medial prefrontal cortex (mPFC) neurons.

**Figure supplement 3.** The augmented generalized integrate-and-fire (aGIF) model accurately predicts the subthreshold voltage and firing patterns of serotonin (5-HT) neurons recorded at room temperature (RT) and 29–30°C.

subthreshold voltage ($R^2$ = 0.128 ± 0.135) and achieving $M_d^*$ scores less than half of those observed in mPFC neurons ($M_d^*$ = 0.352 ± 0.118). This indicates that the passive membrane filter and adaptation mechanisms included in the canonical GIF model were insufficient to capture the behavior of 5-HT neurons. By augmenting the GIF model with our experimentally constrained model of $I_A$, the aGIF model not only better predicted the voltage ($R^2$ = 0.301 ± 0.200, p=1.96e-4; *Figure 3F1*) but also the spike timing ($M_d^*$ = 0.481 ± 0.148, p=0.001; *Figure 3F2*) of 5-HT neurons. While the more general iGIF model exhibited a similar improvement in spike timing predictions over the GIF model ($M_d^*$ = 0.536 ± 0.154, p=5.89e-4), it did not significantly outperform the aGIF model (p=0.644; *Figure 3F2*), suggesting that accounting for additional biophysical mechanisms that regulate spiking beyond those included in the aGIF model would be unlikely to further improve performance. Repeating this process using data collected closer to physiological temperature yielded the same result (*Figure 3—figure supplement 3*). Thus, among the models considered, adding $I_A$ to the subthreshold and spiking mechanisms of the GIF model best accounts for the biophysical mechanisms responsible for shaping the responses of 5-HT neurons to in vivo-like inputs.

Turning to the other main cell type of the DRN, we next analyzed the performance of each model in SOM cells (*Figure 3G*). In these cells, the canonical GIF model produced highly accurate predictions ($R^2$ = 0.600 ± 0.238 and $M_d^*$ = 0.818 ± 0.149), consistent with its high performance previously reported for cortical GABAergic neurons (*Mensi et al., 2012*; *Teeter et al., 2018*). Nonetheless, the iGIF achieved small but significant performance gains ($M_d^*$ = 0.892 ± 0.094, p=0.004 vs. aGIF and p=0.003 vs. GIF; *Figure 3G2*), leading us to select it as our model of SOM neurons.

## Multiple adaptation mechanisms in 5-HT neurons

Our model selection approach identified the most salient components required to capture the input-output functions of individual neurons and allowed us to identify functional differences across cell types. 5-HT neurons were distinguishable from SOM and mPFC cells by their long membrane time constants (*Figure 4A*, *Figure 4—figure supplement 1A*) and by the presence of conspicuously potent and protracted adaptation mechanisms (*Figure 4B–D*). Indeed, in addition to evoking a characteristically large and prolonged adaptation current (*Figure 4C*), action potential firing in 5-HT neurons produced a substantial and long-lasting increase in firing threshold (*Figure 4D*; but note that this effect is somewhat attenuated near physiological temperature, *Figure 4—figure supplement 2*). In contrast, SOM neurons most often displayed either negligible or even depolarizing spike-triggered currents (*Figure 4B and C*) that may underlie the burst firing patterns often observed in this cell type (*Figure 1—figure supplement 2*). These observations derived from the parameters of GIF models are not only consistent with our experimental characterization (*Figures 1–3*), but significantly expand it. Thus, 5-HT neurons are characterized by slow membrane dynamics, $I_A$, and particularly prominent adaptation mechanisms.

## Preferential sensitivity of 5-HT neuron population to the onset of sudden inputs

The development and validation of accurate single-cell models allowed us to identify the population-level computations operating in the DRN. We took advantage of the one-to-one correspondence between our GIF models and real neurons to construct synthetic populations with realistic neuron-to-neuron heterogeneity by sampling from banks of single-cell models (*Figure 5A*). In response to step increases of synaptic-like inputs delivered to the entire population (*Figure 5B* left), the population firing rates (in Hz/neuron; *Figure 5B* right) of 5-HT, SOM, and mPFC neurons (*Figure 5C*) transiently

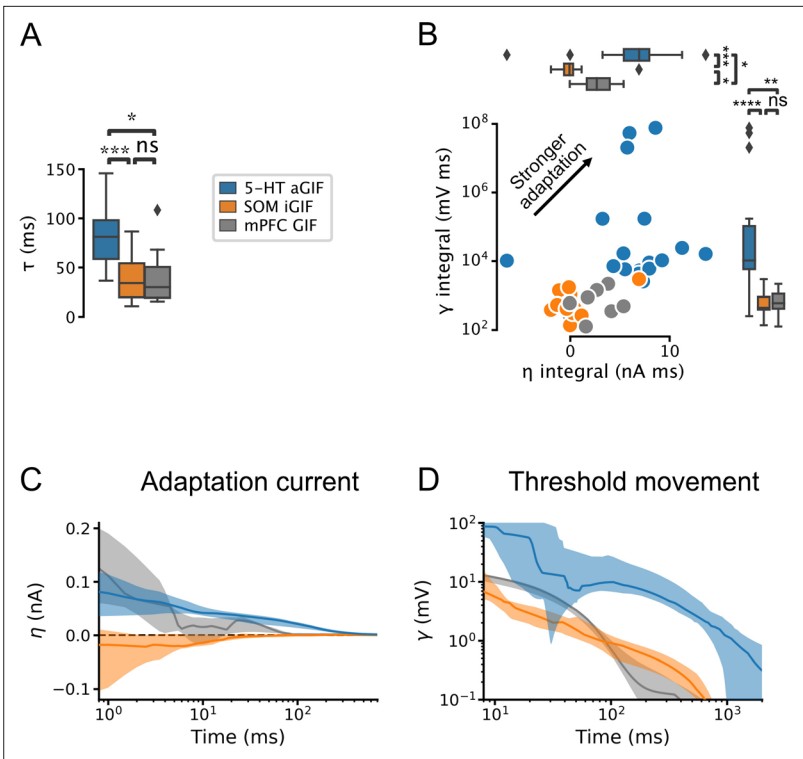

**Figure 4.** Serotonin (5-HT) neurons are distinguished by slow membrane time constants and potent adaptation. (**A**) Using the features from the best performing generalized integrate-and-fire (GIF) model variant for each cell type (legend), passive membrane time constant. (**B**) Spike adaptation features: potency of after-hyperpolarization potential (AHP)-mediated (spike-triggered current $\eta$ integral) and AHP-independent (spike-triggered threshold movement $\gamma$ integral) adaptation. (**C, D**) Comparison of model filters. Presented as median (lines) and interquartile range (bands). Note the long-lasting adaptation currents (C; positive values indicate hyperpolarizing current) and threshold movements (**D**) of 5-HT neurons. Parameters are from models fitted to $N = 18$ 5-HT, $N = 14$ somatostatin (SOM), and $N = 7$ medial prefrontal cortex (mPFC) neurons.

The online version of this article includes the following figure supplement(s) for figure 4:

**Figure supplement 1.** Additional features extracted from single neuron models.

**Figure supplement 2.** Temperature dependence of features extracted from augmented generalized integrate-and-fire (aGIF) models fitted to serotonin (5-HT) neurons.

increased before relaxing to a significantly lower stationary level. Strong inputs did not produce oscillations in the population firing rates, likely because of population heterogeneity (***Figure 5—figure supplement 3***; ***Naud and Gerstner, 2012***; ***Mejias and Longtin, 2012***; ***Mejias et al., 2014***; ***Tripathy et al., 2013***). The transient and stationary parts of the population input-output functions were approximately rectified linear functions (***Figure 5—figure supplement 3***) which we summarized and plotted as the time-varying slope (i.e. gain; ***Figure 5D***). While the gain of the transient response was greater than that of the stationary response in all three cell types, the ratio of transient to stationary gain was substantially higher in 5-HT neurons (***Figure 5E***; ratio of 3.42 ± 0.07 vs. 1.89 ± 0.04 in SOM and 1.50 ± 0.03 in mPFC; p<0.001 in each case; but note that the gain ratio in 5-HT neurons falls to 2.13 ± 0.03 near physiological temperature [***Figure 5—figure supplement 5***], consistent with a smaller spike-triggered threshold movement [***Figure 4—figure supplement 2***]). This marked response of 5-HT cells occurred quickly, in the first 100 ms after the onset of the step. Thus, despite 5-HT neurons being characterized by slow membrane time constants, their population activity provided a remarkably strong encoding of the onset of step synaptic inputs.

We next considered the underlying mechanisms giving rise to the distinctive time-dependent gain of 5-HT neurons. We found that the characteristically strong spike-triggered adaptation of 5-HT neurons (spike-triggered hyperpolarizing adaptation current and threshold movement shown in ***Figure 4***) contributed to the observed relaxation of the population response to a lower stationary

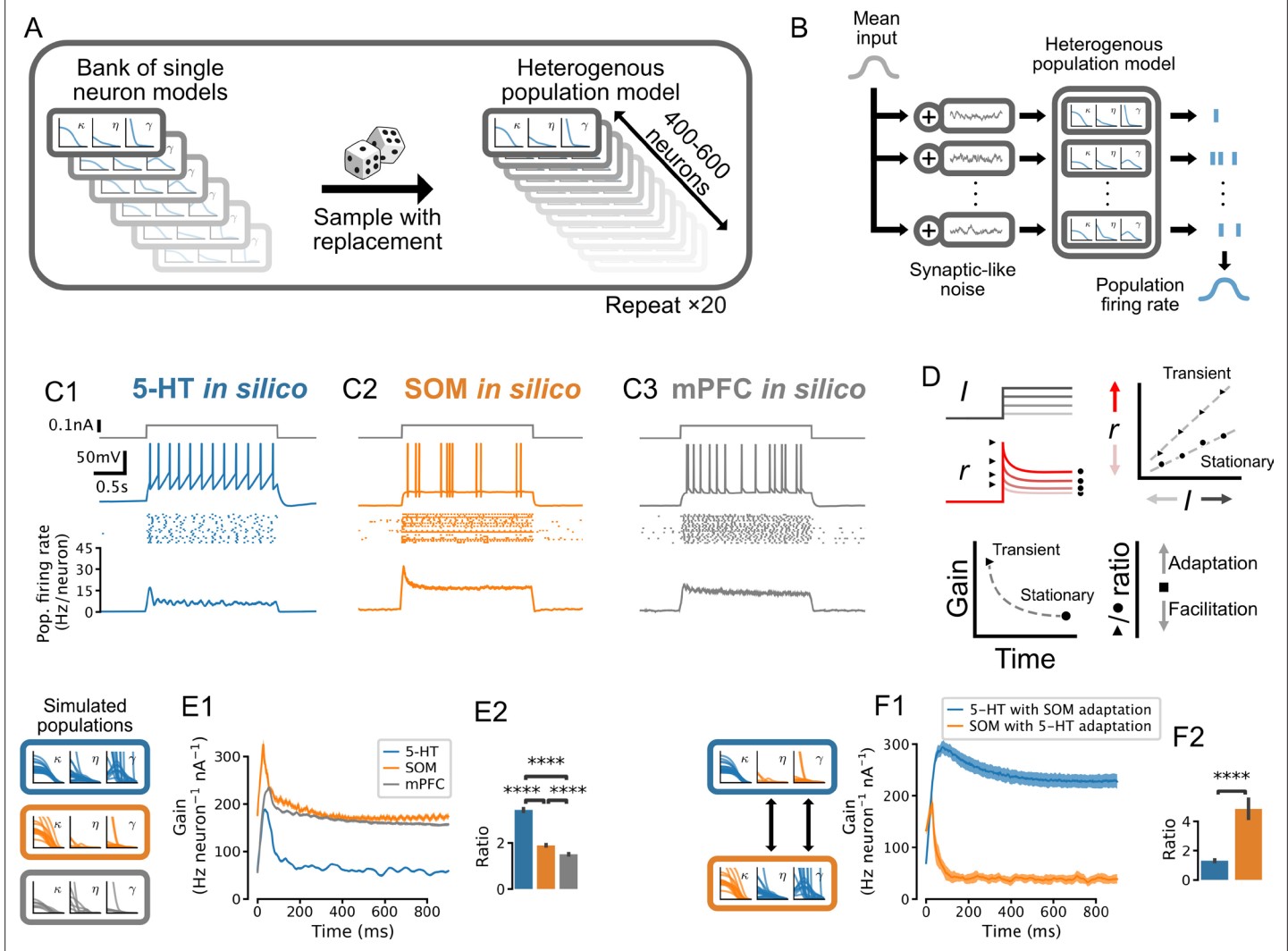

**Figure 5.** Adaptation mechanisms cause a higher gain of the transient vs stationary population response. (**A**) Generation of heterogeneous population models from experimentally constrained single neuron models. (**B**) Schematic of population simulations. Spikes from individual neuron models in the simulated population are added together to produce a population firing rate. (**C**) Population responses to input step. From top in each column: stimulus (gray); sample voltage trace; spike raster of first 20 neurons; mean population firing rate across 20 independent simulations. (**D**) Schematic for quantifying the time-varying population input-output function for both the transient and the stationary components of the response. An input-output function is calculated for the population response at each time point after the input step. The slope of each input-output function (gain) is then plotted as a function of time since the step onset. The ratio of the maximum gain to the minimum gain is a measure of the relative amount of population adaptation. (**E**) Time-resolved gain of step input responses across cell types following the approach shown in D. (**F**) Time-resolved gain of serotonin (5-HT) populations with the adaptation parameters of somatostatin (SOM) neurons (blue) and of SOM populations with adaptation parameters of 5-HT neurons (orange). Data are presented as mean ± SD in E1 and F1. mPFC, medial prefrontal cortex.

The online version of this article includes the following figure supplement(s) for figure 5:

**Figure supplement 1.** Responses to fast and slow inputs are robust to increases in input baseline.

**Figure supplement 2.** Strong responses of neuron populations to sudden inputs are due to a non-linear filtering effect.

**Figure supplement 3.** Simulated population input-output functions across cell types.

**Figure supplement 4.** Characterization of GABAergic synapses on serotonin (5-HT) neurons used to constrain network model.

**Figure supplement 5.** Temperature-dependence of simulated serotonin (5-HT) neuron population gain.

level: grafting the weak adaptation from SOM neuron models onto 5-HT models dramatically reduced the ratio of transient to stationary gain, and vice-versa (*Figure 5F*). These findings are consistent with previous models in other cell types showing that spike-triggered adaptation reduces the sensitivity of neural populations to input changes over long timescales (*Ermentrout, 1998*; *Benda and Herz,*

*2003*; *Naud and Gerstner, 2012*). Therefore the preferential sensitivity of 5-HT neuron populations to sudden changes in synaptic inputs is a natural consequence of strong adaptation at the single neuron level.

## Feedforward inhibition and $I_A$ control 5-HT output gain of the DRN

Apart from the strong adaptation mechanisms of 5-HT neurons, two other mechanisms have the potential to dynamically modulate the 5-HT output from the DRN: $I_A$ in 5-HT neurons and the feed-forward inhibition (FFI) enacted by local DRN interneurons (*Zhou et al., 2017*; *Geddes et al., 2016*). To examine the contributions of these two mechanisms, we first connected our existing SOM population models to 5-HT population models using experimentally constrained GABA$_A$ receptor-mediated synaptic conductances (see Methods and *Figure 5—figure supplement 4*).

To dissect the contribution of $I_A$ in shaping population responses in this connected DRN network, we applied the same inputs to both 5-HT and SOM neuron populations and examined 5-HT neuron population dynamics (as in *Figure 5*) while varying the maximal conductance of $I_A$ (in 5-HT neurons). The gain of the transient component of the 5-HT response increased markedly when the conductance of $I_A$ was set to zero (*Figure 6A*), while increasing the potency of $I_A$ substantially dampened and broadened the population response to fast inputs, reminiscent of $I_A$'s modulation of spike timing jitter observed in our toy model (*Figure 2I–K*). These simulations thus show that $I_A$ substantially regulates the gain of the transient component of DRN 5-HT output evoked by sustained inputs, with negligible effects on the gain of the slower stationary component.

Previous work has shown that glutamatergic excitatory inputs from the PFC make strong mono-synaptic contacts onto both DRN 5-HT and GABAergic neurons, triggering a classic FFI. Intriguingly, the PFC axonal inputs onto these two cellular elements of the DRN are functionally distinct in as much as the PFC synapses onto GABAergic neurons are far more sensitive to endocannabinoid neuromodulation than those onto 5-HT neurons (*Geddes et al., 2016*). The computational role of this differential sensitivity to neuromodulation is currently unknown. We began by determining the role of the DRN FFI per se by comparing the responses of 5-HT neuron population dynamics with or without SOM cells (*Figure 6B*). Including FFI onto 5-HT neurons substantially dampened the overall response of the 5-HT population to synaptic inputs, while still sustaining the preferential encoding of the early phase of sudden inputs (*Figure 6B2*). While introducing FFI did decrease the gain ratio, this decrease was quantitatively smaller than the differences between 5-HT neurons and other cell types shown in *Figure 5E* and the effect of changing $I_A$ shown in *Figure 6A*, *Figure 6B2*. We next directly simulated the effects of endocannabinoid modulation of excitatory input to the DRN observed experimentally (*Geddes et al., 2016*) by weakening the strength of the input to SOM neuron populations by 30% while leaving that to 5-HT neurons intact. By favoring the direct monosynaptic excitation of 5-HT neurons by preferentially diminishing the glutamatergic drive of SOM neurons, this neuromodulation led to an increase in the overall gain of the DRN that was unexpectedly apparent across the entire duration of the response to step inputs (i.e. no change in the gain ratio, *Figure 6B2*). Thus, the target-specific endocannabinoid-mediated modulation of PFC excitatory drive in DRN exerts a normalizing role by increasing the overall gain of 5-HT output evoked by synaptic inputs without altering its preferential encoding of changes in input, which is emerging as a cardinal feature of DRN network dynamics.

Our electrophysiological recordings showed that excitability heterogeneity is a salient feature of the SOM DRN neuron population. Our modeling approach allows us to specifically examine the role of this cellular heterogeneity in shaping the output of the DRN by comparing our DRN model (*Figure 6C*) to an alternative homogenized version in which the parameters of SOM neurons were set to fixed values (*Figure 6D*). Thus, while FFI with an experimentally determined degree of heterogeneity mainly imposed a reduction of the slope of the input-output function (i.e. divisive inhibition), homogeneous FFI mainly shifted the input-output function of the transient component of the population response to the right (i.e. subtractive inhibition; *Figure 6E*). This subtractive feature can be traced back to a strong non-linearity in the input-output functions of homogenized SOM neuron populations (compare *Figure 6E1* and *Figure 6F1*). In the case of the stationary component, both heterogeneous and homogenized DRN models implemented divisive inhibition (*Figure 6F*). Therefore, we conclude that heterogeneity among GABAergic neurons implements divisive inhibition.

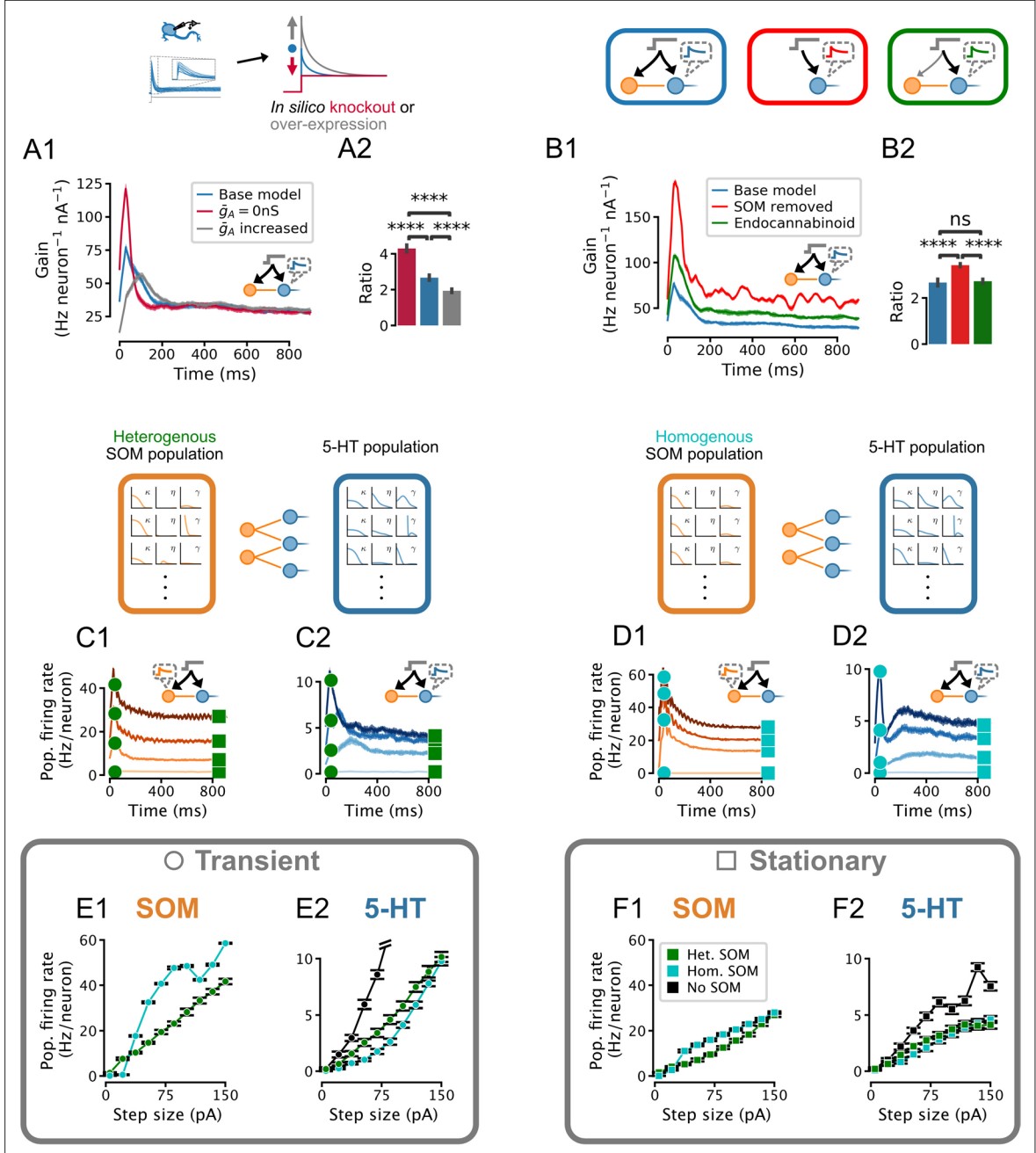

**Figure 6.** Effect of $I_A$ density, feedforward inhibition, and heterogeneity of somatostatin (SOM) neurons on the serotonin (5-HT) neuron population response. Network input is the same set of step stimuli as in **Figure 5D–F**. (**A**) Increasing $I_A$ reduces adaptation by selectively suppressing the early part of the response to sudden inputs, and vice-versa. (**B**) Gain curves with normal feedforward inhibition (blue), with reduced input strength onto the inhibitory population (green), or without inhibition (red). Reduced input strength onto the inhibitory population (green) simulates the effect of endocannabinoid input (**Geddes et al., 2016**). (**A2,B2**) Ratio of peak to steady-state gain. Data are presented as mean ± SD. (**C**) Population firing rates of SOM and 5-HT neurons in a network in which both populations are heterogeneous. (**D**) Population firing rates of SOM and 5-HT neurons in a network in which all SOM neurons are identical. Effects of homogeneous (cyan) or heterogeneous (green) SOM populations on the population input-output functions for the transient (**E**) and stationary (**F**) components of the response (see square and circle markers in C and D). Note that the input-output function of the heterogenous SOM population is approximately linear, whereas that of the homogenous population is not (**E1, F1**). Relative to the input-output functions of a 5-HT population receiving no feed-forward inhibition, the effect of the heterogenous SOM population is divisive, but the effect of the homogenous SOM population on the transient part of the 5-HT population input-output function includes a strong subtractive component (**E2**).

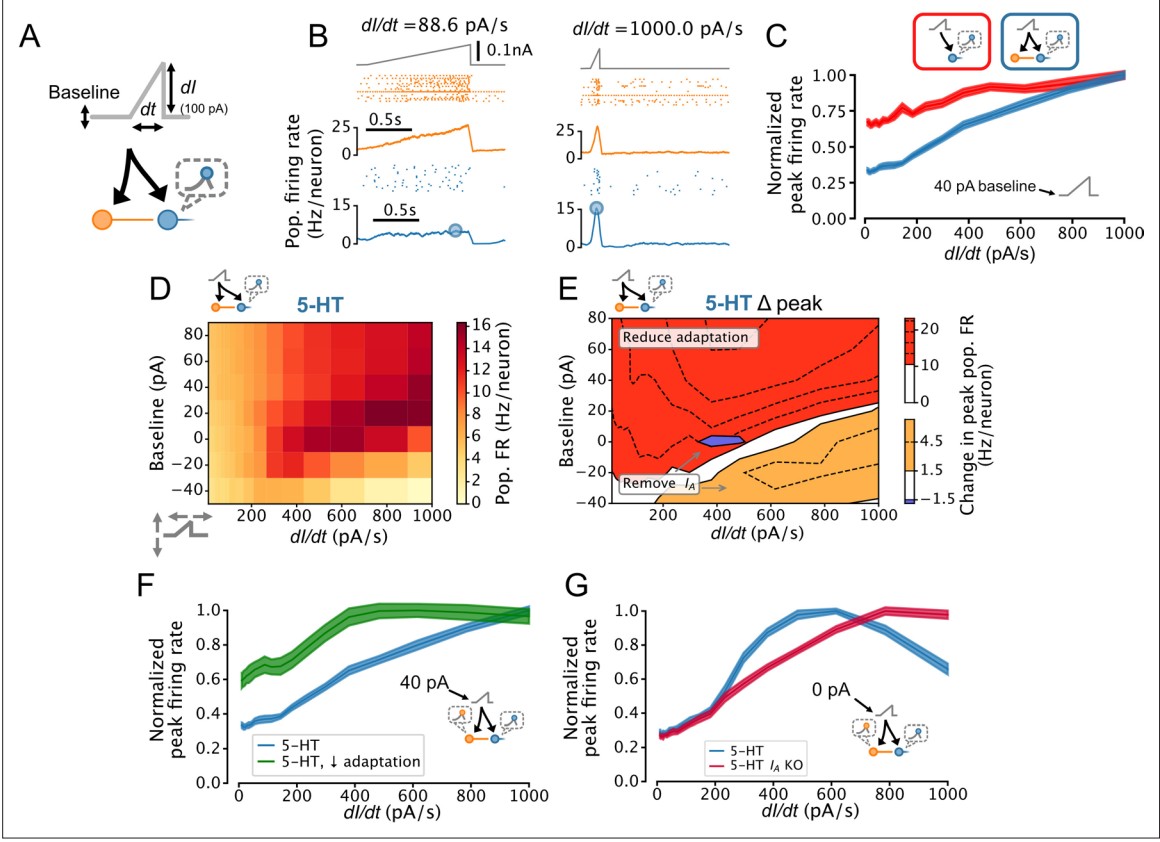

**Figure 7.** Dorsal raphe nucleus (DRN) serotonin (5-HT) neuron population output conditionally encodes the temporal derivative of its input. (**A**) Design of simulations. A ramp stimulus with an adjustable baseline and slope (derivative) is applied to the same network models as in **Figures 5 and 6**, and the peak firing rate (FR) of the 5-HT neuron population is extracted. (**B**) Representative simulated input (top), somatostatin (SOM) neuron population activity (middle), and 5-HT neuron population output (bottom). (**C**) With a baseline input of 40 pA, peak 5-HT neuron population output is approximately linearly related to the derivative of the ramp input, and feed-forward inhibition by SOM neurons enhances this feature. (**D**) Peak FRs of 5-HT neuron populations depend on interacting effects of input baseline and slope. Panel C shows normalized data from the 40 pA row in blue. (**E**) 5-HT neuron adaptation and $I_A$ dominate the DRN input-output function under different input regimes. Effect of reducing 5-HT neuron adaptation (following the approach from **Figure 5F**) is the most pronounced for higher levels of background input and more slowly changing inputs (red), while the effect of removing $I_A$ (following the approach of **Figure 6A**) is the most pronounced for low background input and fast changing inputs (orange, blue). (**F**) Effect of reducing adaptation in 5-HT neuron models visualized at a 40 pA baseline. Note that 5-HT output no longer linearly encodes $dI/dt$ when adaptation is reduced. (**G**) Effect of removing $I_A$ from 5-HT neuron models visualized at a 0 pA baseline. Note that 5-HT output approximately linearly encodes $dI/dt$ when $I_A$ is removed.

The online version of this article includes the following figure supplement(s) for figure 7:

**Figure supplement 1.** Temperature-dependence of temporal derivative encoding by simulated serotonin (5-HT) neuron populations.

**Figure supplement 2.** Toy model of temporal derivative encoding by the dorsal raphe nucleus (DRN).

## 5-HT neurons linearly encode the temporal derivative of inputs to the DRN

Adaptation plays a critical role in implementing temporal derivative encoding in sensory systems (**Lundstrom et al., 2008**; **Pozzorini et al., 2013**) but has not been ascribed a similar role in neuro-modulatory systems such as the DRN. To determine whether the DRN also supports this computation, we parameterized the rate of change of DRN inputs by applying ramp stimuli with variable slopes (i.e. derivatives; **Figure 7A and B**). Remarkably, the peak 5-HT neuron population firing rate linearly reported the slope of the ramps, an effect which was enhanced by FFI (**Figure 7C**). We further found that this linearity was conditional on the presence of slightly depolarizing background input (≥20 pA, **Figure 7D**). Simulations using aGIF models fitted to data collected near physiological temperature yielded similar results; **Figure 7—figure supplement 1**. The potent adaptation mechanisms of 5-HT neurons play a key role in mediating this linear encoding of input derivative, since reducing the

strength of adaptation reduces linearity across a wide range of input baselines (*Figure 7E and F*). Together, these observations suggest that the DRN signals to its brain-wide target a mixture of the intensity and temporal-derivative of its excitatory inputs, and that the derivative-encoding component dominates when the input is increasing rapidly (*Figure 7—figure supplement 2*).

The extent to which the output of the DRN signals the temporal derivative of its input is likely to be limited by several factors, notably: the long membrane time constants of 5-HT neurons (*Table 1*, *Figure 4—figure supplement 1A*), which cause rapidly fluctuating to be filtered out; the fact that firing rates cannot be less than zero, limiting the dynamic range available to encode negative input derivatives; the presence of $I_A$ , which filters out inputs with a high temporal derivative (*Figure 2*); and the level of background input (*Figure 7D*). Because $I_A$ can be partly inactivated by depolarizing background input, the effects of background input and $I_A$ on the derivative-encoding properties of the DRN are expected to interact. Consistent with this idea, removing $I_A$ from 5-HT neurons in our DRN network models extended the range of background input where the peak 5-HT neuron population firing rate is an approximately linear function of the slope of a ramp stimulus (*Figure 7E and G*). In summary, we found that the presence of strong spike-frequency adaptation in 5-HT neurons causes the DRN to signal the rate of change of its input to its brainwide targets, but that this core computation is progressively suppressed when a state of hyperpolarization engages $I_A$ .

## Discussion

Here, we sought to characterize the computational properties of the DRN using a bottom-up approach grounded in experimentally constrained models of the two most abundant cell types in this region: 5-HT and SOM GABA neurons. Consistent with, and extending, previous work, we found that 5-HT neurons were relatively homogeneous and characterized by potent spike-frequency adaptation (*Figure 1*) and by the presence of a strong A-type potassium current (*Figure 2*), while SOM neurons displayed a considerably more heterogeneous excitability profile (*Figure 1* and *Figure 1—figure supplement 2*). Extensions to classical GIF models (*Mensi et al., 2012*; *Pozzorini et al., 2015*) to capture the non-linear subthreshold effects of $I_A$ observed in 5-HT neurons were required to adequately capture the spiking response of 5-HT neurons to naturalistic stimuli (*Figure 3*). This work introduces a new approach to capturing such non-linear subthreshold effects in the form of the aGIF model, which augments the GIF model of *Mensi et al., 2012* with experimentally constrained Hodgkin-Huxley style currents, improving model interpretability without compromising predictive performance. Inspecting the parameters of the best performing GIF models revealed that the substantial spike-frequency adaptation observed in 5-HT neurons is not fully explained by their distinctively large AHPs and is partly mediated by a previously undescribed dynamic spike threshold (*Figure 4*). This model-based approach allowed us to probe causal relationships between specific excitability features and population computations. Thus, we found that the prominent adaptation mechanisms in 5-HT neurons regulated DRN population responses to synaptic inputs (*Figure 5*), that $I_A$ suppressed the response to sudden inputs, and that heterogenous FFI had a divisive rather than subtractive effect on DRN output (*Figure 6*). By further exploring DRN population dynamics, our simulations demonstrated that 5-HT neurons linearly reported a mixture of the intensity and temporal-derivative of their synaptic inputs (*Figure 7*), and that the temporal-derivative dominates DRN output when the input is increasing rapidly (*Figure 7—figure supplement 2*). In summary, this work points to a new computational role for the DRN in encoding the derivative of its inputs and identifies specific cellular and network mechanisms that give rise to this computation and modulate its expression. These results raise important questions about how the selective responses of the DRN to changing synaptic inputs might support its role in guiding animal's behavior in dynamic environments.

### Need for a hybrid biophysical-simplified methodology

The computational and statistical modeling methodology presented here was designed to bridge the gap between specific biophysical mechanisms and network-level computation. Closing this gap has also been the target of complex biophysical simulations, motivated by the hope to create tools for testing disease-related treatments and for untangling the computations performed by large neural networks (*Markram, 2006*; *Billeh et al., 2020*). Preserving the accuracy and identifiability of simpler approaches (*Gerstner and Naud, 2009*; *Mensi et al., 2012*; *Pozzorini et al.,*

*2013*; *Teeter et al., 2018*), the 'augmented GIF' model developed here explicitly incorporates the most important biophysical features of 5-HT neurons, allowing us to probe their contributions to network-level computation by altering or removing the corresponding model components during network simulations. While the aGIF framework was developed here to capture the effects of inactivating subthreshold potassium currents in 5-HT neurons, it lends itself equally well to capturing the effects of other subthreshold voltage-gated currents. We note that, as in other methods based on linear regression of nonlinear ion channel dynamics (*Huys et al., 2006*; *Huys and Paninski, 2009*), adequate experimental estimates of the voltage-dependent gating features of the conductance at play must be available to be inserted in the aGIF model. Altogether, this expanded modeling framework adds to a toolset of computational approaches for interrogating the role of particular microcircuit motifs (e.g. FFI) or excitability features (e.g. spike-triggered adaptation) in shaping network computations, while lending itself to more elaborate inference methodologies (*Gonçalves et al., 2019*).

Could the dynamical features identified here have been captured by a simpler modeling framework? Two closely related approaches that we have not considered here are linear-nonlinear (LNL) and generalized linear models (GLMs), which are trained using only the spike output and external input to each cell and do not consider the subthreshold voltage (*Pillow and Simoncelli, 2006*; *Pillow et al., 2008*). Despite the fact that the GLM approach was not possible here given the very low firing rates of 5-HT neurons and the large number of action potentials required for accurate characterization in the absence of information about the subthreshold voltage, it is worth asking whether GLMs could in principle capture the network-level properties of 5-HT signaling. For instance, the role of spike-triggered adaptation in conveying preferential sensitivity to suddenly changing inputs arises in GLMs (*Naud and Gerstner, 2012*), but the state-dependence of the input derivative sensitivity identified in 5-HT neurons (*Figure 7*) could not have been captured by a GLM implementation. In summary, the GIF framework provides a more solid foundation for network modeling than LNL- or GLM-based approaches for cell types with very low firing rates or highly state-dependent output.

Does the aGIF modeling approach represent an unnecessary complication of the GIF model framework or, conversely, an oversimplification of detailed Hodgkin-Huxley models? GIF models that do not explicitly account for the effects of specific ionic conductances produce highly accurate spiketrain predictions in many cell types (*Gerstner and Naud, 2009*; *Mensi et al., 2012*; *Pozzorini et al., 2013*; *Teeter et al., 2018*); indeed, even in 5-HT neurons, the iGIF model predicts the timing of spikes with an accuracy equal to that of our aGIF model. For questions where the biophysical mechanisms that regulate spiking are not of primary interest and for systems where simpler LNL or GLM models are not able to predict the timing of spikes accurately (e.g. due to low firing rates as discussed above), non-augmented GIF models remain suitable tools. In our case, it would not have been possible to probe the effect of $I_A$ on the network-level processing features of the DRN without the aGIF model.

## Network-level role of $I_A$ current

Previous modeling work has implicated $I_A$ in controlling the sensitivity of the stationary response to sustained inputs (*Connor and Stevens, 1971*; *Connor et al., 1977*; *Tuckwell and Penington, 2014*; *Drion et al., 2015*). These studies contrast with our findings which implicate this current in the control of the transient component but show almost no effect on the stationary component of the response. This discrepancy can be explained by noting that the AHPs of 5-HT neurons (and thus of our computational model) do not reach the hyperpolarized potentials required to free $I_A$ from inactivation (*Figure 1* and *Figure 1—figure supplement 1*), in contrast to the model of *Connor et al., 1977*. As a result, $I_A$ remains mostly inactivated during sustained inputs, and the stationary response is mostly regulated by the interplay between spike-triggered adaptation and the strength of the input. Other factors such as a shift in the activation and/or inactivation curves (e.g. by neuromodulators) are expected to influence how $I_A$ controls the transient and stationary components of the response. Finally, it is interesting to note that $I_A$ is also highly expressed in the dendrites of cortical neurons, where it may have an analogous function (*Hoffman et al., 1997*; *Harnett et al., 2013*; *Ujfalussy et al., 2018*; *Payeur et al., 2019*). Our results hint at a possible general role of $I_A$ in suppressing transient responses to sustained inputs in the midbrain, cortex, and other systems.

## 5-HT neuron heterogeneity

5-HT neurons are not all alike in every respect: recent experimental work has uncovered molecular, electrophysiological (*Calizo et al., 2011*), developmental, and anatomical (*Commons, 2015*; *Ren et al., 2018*) differences among 5-HT neurons across raphe nuclei and within the DRN (reviewed in *Okaty et al., 2019*). Most relevant to our work are previously reported quantitative differences in the excitability of serotonin neurons located in the dorsomedial DRN, ventromedial DRN, and median raphe nucleus (*Calizo et al., 2011*). These observations suggest that the predictions made by our model, which was fitted primarily to serotonin neurons from the ventromedial DRN, may agree qualitatively but not quantitatively with the behavior of 5-HT neuron ensembles in these areas. While there is not yet any evidence that serotonin neurons in different parts of the serotonin system perform qualitatively different computational operations, this remains an intriguing possibility for future work.

## Heterogeneous properties of SOM neurons ensure divisive inhibition

How the heterogeneity of excitability influences the response properties of neuronal populations depends on a number of factors. Specifically, we and others *Mejias and Longtin, 2014* have argued that heterogeneity of feedback inhibition (and of principal cells) implements a divisive effect on the stationary part of the population input-output function. For FFI, a divisive effect on the gain of stationary input-output functions is expected in naturalistic conditions (*Mejias and Longtin, 2014*). The findings outlined here further support these theoretical results by showing that the heterogeneous FFI remains divisive on the transient part of the response. Divisive inhibition has been proposed to be essential to counteract strong excitation so as to maintain activity within an adequate dynamic range (*Chance and Abbott, 2000*; *Ferguson and Cardin, 2020*), and it is expected that brain circuits will harness cellular and circuit-level mechanisms to tune their sensitivity to relevant inputs while maintaining overall stability. This point is germane to 5-HT neurons given their position at the confluence of many excitatory input streams (*Weissbourd et al., 2014*; *Pollak Dorocic et al., 2014*; *Ogawa et al., 2014*; *Zhou et al., 2017*; *Ren et al., 2018*; *Geddes et al., 2016*). Thus, while the exact behavioral function of the 5-HT system is still unclear, uncovering important components of its gain control mechanisms might provide useful hints about how it integrates its multifold inputs.

## Neuromodulation of neuromodulation

Neuromodulators can dynamically reconfigure information processing in neural circuits that are otherwise anatomically fixed (*Marder, 2012*; *Tsuda et al., 2021*). While 5-HT is considered to be a neuromodulator, the DRN network is itself under neuromodulatory influence, both from distal (e.g. locus coeruleus or ventral tegmental area) or local (e.g. endocannabinoids, 5-HT itself) sources (*Baraban and Aghajanian, 1981*; *Aman et al., 2007*; *Weissbourd et al., 2014*; *Geddes et al., 2016*; *Lynn et al., 2022*). Whereas previous work has outlined defined cellular metrics that are modulated by specific receptor subtypes (e.g. changes in release probability or direct membrane depolarization/hyperpolarization), the consequences of these neuromodulatory influences on higher-order network computation are only superficially understood. Here, we showcase two broad neuromodulatory mechanisms that enact different effects on population coding. Through simulations, we show that reducing the magnitude of $I_A$ (which could be caused, for instance, in vivo by noradrenergic input the DRN [*Aghajanian, 1985*]) enhances the sensitivity of the raphe response to the onset of step inputs while leaving the stationary firing rate unchanged. In contrast, the cannabinoid-mediated preferential reduction of FFI onto 5-HT neurons (caused by the tonic activation of DRN endocannabinoid receptors, as expected to occur, for instance, during marijuana recreational or therapeutic use [*Geddes et al., 2016*]) rather causes a general reduction in the output gain of the DRN. Together with our simulations probing the temporal derivative-encoding properties of this region, these observations point to a conceptual model in which the output of the DRN represents a mixture of the intensity and temporal derivative of its input where $I_A$ controls the relative balance of the two components, and FFI regulates the overall intensity of the output, and where these functions can be rapidly and independently tuned by neuromodulatory control.

Our heuristic model of the DRN helps to illustrate the unexpectedly multifaceted nature of the computations performed by this evolutionarily ancient region, but, like most heuristics, it remains an oversimplification. Some of the qualitative features of DRN processing emerging from our simulations are not explained by our "input intensity plus temporal derivative" heuristic (e.g. the ability of FFI to

modulate the temporal derivative-encoding properties of the DRN, or the attenuation of these same coding properties by hyperpolarization [*Figure 7*]), presenting further opportunities to better understand the influence of neuromodulation on network computation in this region.

## Role of derivative encoding in reinforcement learning

The role of 5-HT signaling in modulating behavior is increasingly conceptualized through the lens of reinforcement learning (RL) theory. Indeed, 5-HT output has been proposed to loosely encode or modulate every component of classical RL (*Sutton and Barto, 2018*, *Dayan and Huys, 2009*), including a reward signal (*Li et al., 2016*), state value (*Cohen et al., 2015*; *Luo et al., 2016*), bias in state-action value (*Miyazaki et al., 2018*), temporal discounting factor (*Doya, 2002*; *Schweighofer et al., 2008*), prediction error (*Daw et al., 2002*, but see *Boureau and Dayan, 2011*), and learning rate (*Matias et al., 2017*; *Iigaya et al., 2018*; *Grossman et al., 2022*), with varying degrees of experimental support. Might the derivative-like computation described here have a place in an RL-based conception of DRN function? For now, it is only possible to speculate. Existing RL models of DRN function bin time in increments of tens of seconds, obscuring the faster adaptation dynamics that are the subject of our work. How and whether the sub-second fluctuations in DRN 5-HT neuron activity that are consistently observed in reward learning experiments (*Ranade and Mainen, 2009*; *Cohen et al., 2015*; *Li et al., 2016*; *Zhong et al., 2017*; *Grossman et al., 2022*) should be incorporated into RL models remains unclear. Our results suggest that RL operations that can be seen as computing a temporal derivative are candidates for an RL-based account of DRN function.

If the electrophysiological features of individual 5-HT neurons directly participate in shaping the computations enacted by the DRN, the same is likely true for other neuromodulatory systems, and this work may offer overall guiding principles. For instance, dopamine neurons, well known for their reward prediction error-like coding properties (*Schultz et al., 1997*), bear some electrophysiological features in common with DRN 5-HT neurons, with both cell types exhibiting strong adaptation and a prominent A-type potassium current (*Grace and Onn, 1989*, *Khaliq and Bean, 2008*). Dopamine neurons have been proposed to encode reward prediction errors partly by approximating a mixture of a value signal and its temporal derivative (*Kim et al., 2020*), hinting at a possible role for adaptation in implementing one of the central computations of RL.

If the derivative-like operation identified here does not directly contribute to computing one of the key components of RL, what might its role in the DRN be? One possibility is that strong spike-triggered adaptation may optimize the efficiency of neural coding by filtering out temporally redundant information, a phenomenon referred to as predictive coding and that is ubiquitous in sensory systems (*Brenner et al., 2000*; *Barlow, 2001*; *Ulanovsky et al., 2003*; *Kohn, 2007*). As the search for a unified interpretation of DRN 5-HT activity continues, our results provide a new perspective on the fast component of 5-HT neuron dynamics: fluctuations in 5-HT neuron's activity do not solely encode the intensity of their input, but rather how quickly their inputs are changing over time.

## Materials and methods
### Experimental methods
#### Animals

Experiments were performed on male and female C57/Bl6 mice aged 4–8 weeks. *Slc6a4*-cre::Rosa-TdTomato (SERT-Cre) and *Sst*-cre::Rosa-TdTomato transgenic lines were used to fluorescently label DRN 5-HT and somatostatin (SOM) GABA neurons, respectively. Animals were group-housed and kept on a 12:12 hr light/dark cycle with access to food and water ad libitum. All experiments were carried out in accordance with procedures approved by the University of Ottawa Animal Care and Veterinary Services (protocol numbers CMM-164, CMM-176, CMM-1711, CMM-1743, and CMM-2737).

#### Slice preparation

Animals were deeply anesthetized using isofluorane (Baxter Corporation) before being euthanized by decapitation. The brain was quickly removed from the skull and submerged into ice-cold dissection buffer containing the following: 119.0 mM choline chloride, 2.5 mM KCl, 4.3 mM $MgSO_4$, 1.0 mM $CaCl_2$, 1.0 mM $NaH_2PO_4$, 1.3 mM sodium ascorbate, 11.0 mM glucose, 26.2 mM $NaHCO_3$; saturated with 95% $O_2$/5% $CO_2$. A Leica VT1000S vibratome was used to cut 300-μm coronal sections

of midbrain containing the DRN or of the cortex containing the mPFC in the same ice-cold choline dissection buffer. After cutting, slices were placed in a recovery chamber filled with artificial cerebrospinal fluid containing the following: 119.0 mM NaCl, 2.5 mM KCl, 1.3 mM MgSO$_4$, 2.5 mM CaCl$_2$, 1.0 mM NaH$_2$PO$_4$, 11.0 mM glucose, 26.2 mM NaHCO$_3$; ~298 mOsm, maintained at 37°C, and continuously bubbled with 95% O$_2$/5% CO$_2$. The recovery chamber was allowed to equilibrate to room temperature for 1 h before beginning experiments.

## In vitro whole-cell electrophysiological recording

Neurons were visualized using an upright microscope (Olympus BX51WI) equipped with differential interference contrast and a ×40, 0.8 NA water-immersion objective. Whole-cell recordings were obtained from fluorescently labeled DRN 5-HT and SOM neurons and unlabeled mPFC L5 pyramidal neurons using glass electrodes (Sutter Instruments; tip resistance 4–6 MOhm). For most experiments, the following potassium gluconate-based internal solution was used: 135 mM potassium gluconate, 6.98 mM KCl, 10 mM HEPES, 4 mM Mg ATP, 0.40 mM GTP, 10 mM Na phosphocreatine; adjusted to pH 7.25 with KOH, 280–290 mOsm. A subset of experiments (GABA synaptic physiology) were carried out using a cesium-based internal solution (120 mM CsMeSO$_3$, 10 mM EGTA, 5 mM TEA Cl, 1 mM CaCl$_2$, 10 mM Na HEPES, 4 mM Mg ATP, 2 mM GTP, 2 mM QX-314, and 10 mM Na phosphocreatine; adjusted to pH 7.25 with CsOH, 280–290 mOsm) and in the presence of bath-applied 100 µM (2 R)-amino-5-phosphonovaleric acid (APV) and 5 µM 2,3-dioxo-6-nitro-1,2,3,4-tetrahydrobenzo[f] quinoxaline-7-sulfonamide (NBQX). For voltage clamp experiments, whole-cell capacitance compensation was applied manually following break-in, and leak current subtraction was performed post hoc using membrane leak conductance estimated based on a –5 mV pulse at the start of each sweep. Experiments were carried out at room temperature except where noted. For current clamp experiments used to fit GIF models, access resistance was compensated using an active electrode compensation method (*Pozzorini et al., 2015*). For voltage clamp experiments used to characterize $I_A$ in 5-HT neurons at room temperature, recordings had $R_a$ = 14.7 ± 6.2 MOhm (mean ± SD; half of recordings between 12.8 MOhm and 21.6 MOhm) after applying an access resistance cutoff of 30 MOhm (a more stringent cutoff of 20 MOhm yielded statistically indistinguishable estimates of $I_A$ maximal conductance and kinetic parameters; compare *Figure 1D* and *Figure 1—figure supplement 5*). For voltage clamp experiments used to characterize whole-cell currents in SOM neurons, recordings had $R_a$ = 14.3 ± 7.0 MOhm (mean ± SD; half of recordings between 9.8 MOhm and 15.5 MOhm) after applying a similar cutoff of 30 MOhm. For synaptic electrophysiology experiments, recordings had $R_a$ = 5.7 ± 0.5 MOhm (mean ± SD; range 5.0 MOhm–6.1 MOhm) after applying a cutoff of 10 MOhm. Recordings were collected with an Axon MultiClamp 700B amplifier, and the analog signals were filtered at 2 kHz and digitized at 10 kHz using an Axon Digidata 1550 digitizer.

## Models

### GIF and related models

The GIF and Na-inactivation GIF (iGIF) models have been described previously in detail (*Mensi et al., 2012*; *Pozzorini et al., 2015*; *Mensi et al., 2016*). Briefly, the GIF and iGIF are composed of a subthreshold component which integrates input currents into voltage and a stochastic spiking rule which transforms subthreshold voltage into a series of spikes. The subthreshold dynamics of the GIF and iGIF are given by

$$C\frac{dV}{dt} = -g_l(V(t) - E_l) - \sum_{\hat{t}_i < t} \eta(t - \hat{t}_i) + I_{\text{inj}}(t) \tag{1}$$

where $\{\hat{t}_i\}$ is the set of spike times and $\eta(t) = \sum_j w_j \exp\left[-t/\tau_j^{(\eta)}\right]$ is the spike-triggered adaptation current. Here the $w_j$ are coefficients estimated from the data and the $\tau_j^{(\eta)}$ are fixed hyperparameters; see Appendix for details. The GIF emits spikes according to an inhomogeneous Poisson process with intensity $\lambda(t)$, given by

$$\lambda(t) = \lambda_0 \exp\left[\frac{V(t) - V_T^* - \sum_{\hat{t}_i < t} \gamma(t - \hat{t}_i)}{\Delta V}\right] \tag{2}$$

where $V_T^*$ is the stationary threshold, $\gamma(t) = \sum_j \beta_j^{(\gamma)} \exp\left[-t/\tau_j^{(\gamma)}\right]$ is the spike-triggered threshold movement (where the $\beta_j^{(\gamma)}$ are coefficients estimated from the data and the $\tau_j^{(\gamma)}$ are fixed; see Appendix), $\Delta V$ is the threshold sharpness (mV; larger values increase the stochasticity of spiking), and $\lambda_0 = 1$ Hz is a constant such that $\lambda(t)$ is in units of Hz. In the iGIF, an additional variable $\theta(t)$ is added to the numerator of the exponentiated term in *Equation 2* to account for voltage-dependent changes in threshold:

$$\lambda(t) = \lambda_0 \exp\left[\frac{V(t) - V_T^* - \sum_{\hat{t}_i < t} \gamma(t - \hat{t}_i) + \theta(t)}{\Delta V}\right]$$

$$\frac{d\theta}{dt} = \frac{\theta_\infty(V) - \theta}{\tau^{(\theta)}}.$$

The equilibrium voltage-dependent change in spike threshold $\theta_\infty(V) = \sum_{j=1}^{N_{step}} \beta_j^{(\theta)} \text{rect}\left[V; A_j, A_{j+1}\right]$ is a piecewise constant function of voltage where each $\beta_j^{(\theta)}$ defines the value of $\theta_\infty(V)$ over the voltage range $[A_j, A_{j+1})$ and $N_{step} = 5$. The locations of the steps in the piecewise constant function $A_j$ are selected based on the data. (See *Mensi et al., 2016* for details on the iGIF model.) Our aGIF model is identical to the GIF model except that two Hodgkin-Huxley currents which together capture the voltage-gated potassium currents found in 5-HT neurons (see 'Potassium current', below) are added to the subthreshold dynamics given in *Equation 1*, yielding

$$C\frac{dV}{dt} = -g_l\left(V(t) - E_l\right) - I_A(t) - I_K(t) - \sum_{\hat{t}_i < t} \eta\left(t - \hat{t}_i\right) + I_{inj}(t) \tag{3}$$

as the definition of the subthreshold dynamics of the aGIF model.

The procedures for fitting the GIF and iGIF models to electrophysiological data have also been described previously in detail (*Mensi et al., 2012*; *Pozzorini et al., 2015*; *Mensi et al., 2016*). Briefly, parameter estimation for both models occurs in two stages: first, the subthreshold parameters are estimated by regression, and second, the threshold parameters are estimated by maximizing the likelihood of the observed spiketrain as a function of the threshold parameters. The fitting procedure for the aGIF is very similar to that of the GIF, with adjustments to the subthreshold fitting procedure to accommodate the extra terms in *Equation 3* (see Appendix for details). Neurons with non-stationary firing statistics (Pearson correlation between number of spikes and validation sweep number above 0.9) or highly variable spike timing (intrinsic reliability <0.1) were automatically excluded from our analysis. Exclusion criteria were fixed before comparing candidate models.

## LIF neuron with an inactivating potassium current

Our toy model of a neuron with an inactivating potassium current is based on an LIF augmented with $I_A(t)$ (see 'Potassium current' below):

$$C\frac{dV}{dt} = -g_l\left(V(t) - E_l\right) - I_A(t) + I_{inj}(t),$$

where $g_l$ and $E_l$ are the leak conductance and reversal, respectively, and $I_{inj}(t)$ is the external input to the model. To reduce the number of free parameters, the model we used is non-dimensionalized with respect to the membrane time constant $\tau_{mem} = C/g_l$ and leak conductance $g_l$, yielding

$$\frac{dV}{dt} = E_l - V(t) - \bar{g}_A' m_\infty(t) h(t)\left(V(t) - E_K\right) + V_{inj}(t)$$

where $t$ is in units of the membrane time constant, $\bar{g}_A' = \bar{g}_A/g_l$ is the effective maximum conductance associated with $I_A$, and $V_{inj}(t) = I_{inj}(t)/g_l$ is the effective external input. The gating variables $m_\infty$ and $h$ are described below in 'Potassium current'.

## Potassium current

The voltage-gated potassium currents in 5-HT neurons were modeled in terms of an inactivating current and a non-inactivating current, we refer to as $I_A$ and $I_K$, respectively. These were defined as follows

$$I_A = \bar{g}_A \, m_\infty(V) \, h(t) \, (V(t) - E_K)$$
$$I_K = \bar{g}_K \, n_\infty(V) \, (V(t) - E_K)$$

(4)

where $\bar{g}$ is the maximal conductance; $m$ and $h$ are the activation and inactivation gates of $I_A$, respectively; $n$ is the activation gate of $I_K$; and $E_K = -101$ mV is the reversal potential of potassium in our recording conditions. Note that although this value is not physiological, the effect of varying this parameter is very similar to the effect of varying $\bar{g}_A$, as we have done in the result section. For simulations involving models fitted to data collected at 29–30°C, $E_K = -89.1$ mV was used. The equilibrium state of each gate $x \in \{m, h, n\}$ is a sigmoid function of voltage

$$x_\infty(V) = \frac{A_x}{1 + e^{-k_x(V - V_x^*)}}$$

where $V_x^*$ is the half-activation voltage (mV), $k_x$ is the slope (mV$^{-1}$), and $A_x$ is a scaling factor.

To keep the number of parameters in our current model to a minimum, we assumed that the $m$ and $n$ gates have instantaneous kinetics (allowing their corresponding equilibrium gating functions $m_\infty$ and $n_\infty$ to be used directly in *Equation 4*), and that the $h$ gate inactivates and de-inactivates with a single time constant $\tau_h$ (ms) that does not depend on voltage. The time dynamics of the $h$ gate are therefore given by

$$\frac{dh}{dt} = \frac{h_\infty - h}{\tau_h}.$$

## Quantification of single-neuron model performance

$R^2$ was calculated based on the training set $\frac{dV}{dt}$ predicted by the subthreshold component of a given GIF model (*Equations 1 and 3*, where the spike times $t$ were constrained to match the data), excluding a small window around each spike (from 1.5 ms before to 6.5 ms after in 5-HT neurons, and from 1.5 ms before to 4.0 ms after in SOM and mPFC neurons). $M_d^*$ was calculated based on validation set data as previously described by *Naud et al., 2011*. This metric is defined as

$$M_d^* = \frac{2 n_{dm}}{n_{dd}^* + n_{mm}},$$

where $n_{dm}$ is the number of model-predicted spikes that occur within 8 ms of a spike in the validation data, and $n_{dd}^*$ and $n_{mm}$ are the corresponding numbers of coincident spikes across sweeps in the validation data and model predictions (where $n_{dd}^*$ is corrected for small sample bias). $M_d^*$ can be interpreted as the fraction of model-predicted spikes that occur within 8 ms of a spike emitted by a real neuron (the spike timing precision is set to 8 ms by inspecting the relationship between precision and intrinsic reliability [*Jolivet et al., 2008*]), corrected such that the chance level is 0 and perfect agreement between predicted and observed spikes is 1.

## Population models

DRN network models were constructed by connecting a population of 400 SOM neuron models to a population of 600 5-HT neuron models in a feed-forward arrangement. Population models were bootstrapped by sampling with replacement from a bank of experimentally constrained GIF models. SOM neuron models were randomly connected to 5-HT neuron models with a connection probability of 2%, such that the expected number of GABAergic synapses on each 5-HT neuron model was 8. We used a conductance-based model of GABAergic synapses with a fixed reversal potential of –76.7 mV, conductance of 0.3 nS, and biexponential kinetics with $\tau_{rise} = 1.44$ ms, $\tau_{decay} = 26.0$ ms, and a propagation delay of 2.0 ms.

Simulated 5-HT populations with decreased or increased $I_A$ were generated by setting $\bar{g}_A$ in all single neuron models to 0 nS or 10 nS, respectively. DRN network models with homogenized SOM neuron populations were created by setting all SOM neuron model parameters to their respective median values from the bank of experimentally constrained single neuron models. Population models in which the adaptation mechanisms of 5-HT and SOM neuron models were swapped were generated by randomly sampling a GIF model of the opposite cell type and substituting in its adaptation filter coefficients $\beta_j^{(\gamma)}$ and $w_j$. This procedure is summarized in the following pseudocode:

```
for 5-HT_model in 5-HT_population; do
```

```
SOM_model = random_choice(SOM_models)
5-HT_model.eta.coefficients= SOM_model.eta.coefficients
5-HT_model.gamma.coefficients= SOM_model.gamma.coefficients
end for.
```

## Numerical methods

Simulations were implemented in Python and C++ using custom-written extensions of the GIF Fitting Toolbox (*Pozzorini et al., 2015*; original code archived at https://github.com/pozzorin/GIFFitting-Toolbox; *Pozzorini, 2016*). Numerical integration was performed using the Euler method with a time step of 0.1 ms for the GIF model and related models (to match the sampling rate of electrophysiological recordings) and $0.001\ \tau_{mem}$ for the toy model of a neuron with $I_A$.

## Statistics

Statistical analysis was carried out using the SciPy and statannot (https://github.com/webermar-colivier/statannot; *Weber, 2022*) Python packages. Non-parametric tests were used for all two-sample comparisons (Mann-Whitney U test for unpaired samples and Wilcoxon signed-rank test for paired samples). Non-parametric tests were chosen because we often had reason to believe that our data did not come from a normal distribution, either due to intrinsic qualities of the data, such as being bounded between 0 and 1, or due to skewness apparent in our samples. Whenever multiple tests were performed in the same figure panel, p-values were adjusted for multiple comparisons using the Bonferroni correction. '*', '**', '***', and '****' are used in figures to denote statistical significance at the $p \leq 0.05$, 0.01, 0.001, and 0.0001 levels, respectively, and 'o' is used to indicate a trend toward significance (defined as $0.05 < p \leq 0.1$). Exact p-values are reported in the main text, and summary statistics are presented as mean ± SD. Sample sizes always refer to biological replicates.

## Acknowledgements

This work was carried out on the unceded and unsurrendered land of the Algonquin Anishinaabe people. We thank Dr. Simon Chen for providing SOM-Cre::Rosa-TdTomato mice and Liwen Cai and David Lemelin for technical assistance with mouse lines. We would also like to thank Sébastien Maillé for contributions to pilot-testing of the GIF model in 5-HT neurons and all other members of the J-CB. and RN labs for helpful discussions. EH is thankful to have received graduate scholarships from the Canadian Institutes of Health Research and the Natural Sciences and Engineering Research Council of Canada. This work was supported by grants from the Canadian Institutes of Health Research, the Natural Sciences and Engineering Research Council of Canada, the Canada Foundation for Innovation, Brain Canada (Canadian Neurophotonic Platform), and the Krembil Foundation.

## Additional information

### Funding

| Funder | Grant reference number | Author |
|---|---|---|
| Canadian Institutes of Health Research | 175325 | Richard Naud<br>Jean-Claude Béïque |
| Canadian Institutes of Health Research | 175319 | Richard Naud<br>Jean-Claude Béïque |
| Natural Sciences and Engineering Research Council of Canada | 06972 | Richard Naud |
| Canada Foundation for Innovation, Brain Canada (Canadian Neurophotonic Platform) | | Jean-Claude Béïque |
| Krembil Foundation | | Jean-Claude Béïque |

| Funder | Grant reference number | Author |
| --- | --- | --- |

The funders had no role in study design, data collection and interpretation, or the decision to submit the work for publication.

## Author contributions

Emerson F Harkin, Conceptualization, Data curation, Software, Formal analysis, Validation, Investigation, Visualization, Methodology, Writing - original draft, Project administration, Writing – review and editing; Michael B Lynn, Conceptualization, Investigation, Methodology; Alexandre Payeur, Conceptualization, Formal analysis, Validation, Investigation, Methodology; Jean-François Boucher, Data curation, Investigation; Léa Caya-Bissonnette, Dominic Cyr, Chloe Stewart, Investigation; André Longtin, Conceptualization, Writing – review and editing; Richard Naud, Conceptualization, Resources, Formal analysis, Supervision, Funding acquisition, Validation, Methodology, Writing – review and editing; Jean-Claude Béïque, Conceptualization, Resources, Supervision, Funding acquisition, Validation, Writing – review and editing

## Author ORCIDs

Emerson F Harkin http://orcid.org/0000-0003-0698-5894
Michael B Lynn http://orcid.org/0000-0003-0760-4555
Alexandre Payeur http://orcid.org/0000-0002-2437-8249
Léa Caya-Bissonnette http://orcid.org/0000-0002-2893-6949
André Longtin http://orcid.org/0000-0003-0678-9893
Richard Naud http://orcid.org/0000-0001-7383-3095
Jean-Claude Béïque http://orcid.org/0000-0001-7278-4906

## Ethics

All experiments were carried out in accordance with procedures approved by the University of Ottawa Animal Care and Veterinary Services (protocol numbers CMM-164, CMM-176, CMM-1711, CMM-1743, and CMM-2737). At the beginning of each experiment, animals were deeply anesthetized using isofluorane to minimize suffering before being euthanized.

## Decision letter and Author response

Decision letter https://doi.org/10.7554/eLife.72951.sa1
Author response https://doi.org/10.7554/eLife.72951.sa2

# Additional files

## Supplementary files

• MDAR checklist

## Data availability

Raw data is available on Dryad at https://doi.org/10.5061/dryad.66t1g1k2w. Code to fit models, run simulations, and reproduce figures is available at https://github.com/nauralcodinglab/raphegif, (copy archived at swh:1:rev:0a11ab4fe19fa54ddb3f734ad9131d6789b6bed5).

The following previously published dataset was used:

| Author(s) | Year | Dataset title | Dataset URL | Database and Identifier |
| --- | --- | --- | --- | --- |
| Harkin EF, Lynn M, Boucher J, Caya-Bissonnette L, Cyr D, Stewart C, Béïque J | 2021 | Patch-clamp recordings from dorsal raphe neurons | https://dx.doi.org/10.5061/dryad.66t1g1k2w | Dryad Digital Repository, 10.5061/dryad.66t1g1k2w |

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

## Appendix 1

### Definition of aGIF model subthreshold dynamics

The subthreshold dynamics of the aGIF model are given by

$$C\frac{dV}{dt} = -g_l(V(t) - E_l) - \bar{g}_A m_\infty(V(t))h(t)D_K(V(t)) - \bar{g}_K n_\infty(V(t))D_K(V(t)) - \sum_{\hat{t}_i < t} \eta(t - \hat{t}_i) + I_{\text{inj}}(t), \quad \text{(A1)}$$

where $V(t)$ is the membrane voltage, $I_{\text{inj}}(t)$ is an externally applied current, $D_K = V(t) - E_K$ is the driving force on potassium, and $\sum_{\hat{t}_i < t} \eta(t - \hat{t}_i)$ is the adaptation current $\eta$ summed over all past spikes $\{\hat{t}_i \in \mathcal{S} : \hat{t}_i < t\}$ ($\{\mathcal{S}\}$ is the set of all spike times).

The adaptation current produced by a single spike $\eta(t - \hat{t}_i)$ is implemented as a sum of $k$ exponentials given by

$$\eta(t - \hat{t}_i) = \begin{cases} \sum_{j=1}^k w_j e^{-(t - \hat{t}_i)/\tau_j} & \text{if } t > \hat{t}_i \\ 0 & \text{otherwise,} \end{cases} \quad \text{(A2)}$$

where the timescales $\tau_1, \tau_2, ..., \tau_k$ are treated as hyperparameters. If we substituted this implementation of $\eta$ back into **Equation A1**, the term associated with the spike-triggered current $\sum_{\hat{t}_i < t} \eta(t - \hat{t}_i)$ would become $\sum_{\hat{t}_i < t} \sum_{j=1}^k w_j e^{-(t - \hat{t}_i)/\tau_j}$. This double sum can be written more concisely as

$$\sum_{\hat{t}_i < t} \eta(t - \hat{t}_i) = \sum_{j=1}^k w_j \hat{\eta}_j(t), \quad \text{(A3)}$$

where $\hat{\eta}_j(t) = \sum_{\hat{t}_i < t} e^{-(t - \hat{t}_i)/\tau_j}$ is a basis for the adaptation current over the timescale $\tau_j$.

Substituting the definition of the adaptation current from **Equation A3** into **Equation A1**, we obtain a detailed definition of the aGIF model subthreshold dynamics as follows:

$$\frac{dV}{dt} = \frac{1}{C}\Big( -g_l(V(t) - E_l) - \bar{g}_A m_\infty(t)h(t)D_K(t) - \bar{g}_K n_\infty(t)D_K(t) - w_1 \hat{\eta}_1(t) - \cdots$$
$$- w_k \hat{\eta}_k(t) + I_{\text{inj}}(t) \Big). \quad \text{(A4)}$$

### Estimating aGIF model subthreshold parameters

Given a training dataset $\mathcal{D} = \{(V(t), I_{\text{inj}}(t)) : 1 \leq t \leq T\}$, knowledge of the equilibrium gating functions $m_\infty, h_\infty,$ and $n_\infty$, and appropriate choices of $\tau_1, \tau_2, ..., \tau_k$ in $\eta$, our goal is to estimate the remaining parameters in **Equation A4**; namely, $g_l, C, E_l, \bar{g}_A, \bar{g}_K, w_1, ..., w_k,$ and $\tau_h$, where $\tau_h$ is the time constant of the inactivation gate $h$. Fortunately, all of these except for $\tau_h$ can be estimated easily using linear regression.

We begin by rewriting **Equation A4** as the product of a row vector of predictors $\mathbf{x}$ and a column vector of coefficients $\beta$ as follows

$$\frac{dV}{dt} = \mathbf{x} \cdot \beta \quad \text{(A5)}$$

$$= \begin{bmatrix} V(t) \\ 1 \\ m_\infty h D_K \\ n_\infty D_K \\ \hat{\eta}_1(t) \\ \vdots \\ \hat{\eta}_k(t) \\ I_{\text{inj}}(t) \end{bmatrix}^\top \cdot \begin{bmatrix} -g_l/C \\ g_l E_l/C \\ -\bar{g}_A/C \\ -\bar{g}_K/C \\ -w_1/C \\ \vdots \\ -w_k/C \\ 1/C \end{bmatrix}. \quad \text{(A6)}$$

We solve this subject to $g_l, C, \bar{g}_A, \bar{g}_K \geq 0$ using `scipy.optimize.lsq_linear`.

Next we turn to the question of calculating all of the components of $\mathbf{x}$. Because *Equation A4* only reflects the subthreshold dynamics of the aGIF model, we begin by removing all time points in $\mathcal{D}$ within a small window around each spike (from 1.5 ms before each spike until the end of the absolute refractory period). Given the voltages in the cleaned dataset and the set of spike times, it is simple to calculate $m_\infty, h_\infty, n_\infty, D_K$ and $\hat{\eta}_i(t)$. To calculate $h$ for each time point, we order the values of $h_\infty$ according to time and integrate $h$ numerically using a fixed time step, and the initial condition $h = h_\infty$. This has the effect of assuming that the dynamics of the $h$ gate are paused just before each spike and resumed at the end of the refractory period.

The variance explained by the subthreshold model is a non-convex function of $\tau_h$. We therefore conducted a line search over plausible values of $\tau_h$ and chose the value associated with the highest variance explained. This is equivalent to solving

$$\arg \min_\theta \left\| \widehat{\frac{dV}{dt}} - \frac{dV}{dt} \right\| \tag{A7}$$

where $\theta = (\beta, \tau_h)$.

Single-neuron model hyperparameters. For more details on the iGIF model hyperparameter , see *Mensi et al., 2016*.

| Model | Parameter | Symbol | Cell type | Value (ms) |
|---|---|---|---|---|
| All | $\eta$ timescales | $\tau_1, \tau_2, ..., \tau_k$ | All | 3, 10, 30, 100, 300, 1000, 3000 |
| All | $\gamma$ timescales | None | All | 3, 30, 300, 3000 |
| | | | 5-HT | 6.5 |
| All | Refractory period | None | SOM and mPFC | 4.0 |
| iGIF | Candidate threshold-coupling timescales | $\tau_\theta$ | All | 1, 2, 5, 10, 22, 46, 100 |
| aGIF | Candidate inactivation timescales | $\tau_h$ | All | 10, 13, 18, 25, 33, 45, 61, 82, 111, 150 |

