## [Editor Report]

To characterize physiological properties of dorsal raphe serotonin neurons, the authors applied the approach called an augmented generalized integrate-and-fire [aGIF] model, which incorporates a relatively small number of salient biophysical properties of a specific neuron type, and whose parameters are optimized based on voltage dynamics obtained experimentally. The results showed that after-hyperpolarization and A-type potassium currents, in combination with heterogeneous feedforward inhibition from local GABA neurons, give rise to a derivative-like input-output relationship in serotonin neurons.

---

## [Decision Letter]

**Decision letter after peer review:**

Thank you for submitting your article "Temporal derivative computation in the dorsal raphe network revealed by an experimentally-driven augmented integrate-and-fire modeling framework" for consideration by *eLife*. Your article has been reviewed by 3 peer reviewers, and the evaluation has been overseen by a Reviewing Editor and John Huguenard as the Senior Editor. The following individuals involved in review of your submission have agreed to reveal their identity: Jochen Roeper (Reviewer #1); Hitoshi Morikawa (Reviewer #2); Paul Miller (Reviewer #3).

Essential revisions:

In this study, Harkin and colleagues examined the biophysical properties of dorsal raphe (DR) serotonin neurons using a combination of whole-cell patch clamp recording in brain slices and biophysical computational modeling. The authors adopted the approach called a generalized integrate-and-fire [aGIF] model, which incorporates a relatively small number of salient biophysical properties (ionic currents) of a specific neuron, and whose parameters are optimized based on voltage dynamics obtained experimentally. First, the authors observe two prominent features in serotonin neurons. The first is a prominent spike frequency adaptation, resulting from after-hyperpolarization potentials. The second is a "kink" in the voltage trace leading up to the first spike, which can be attributed to the effect of A-type potassium currents (I-A). Using the data, the authors constructed aGIF model of serotonin neurons which contains I-A, two Hodgkin-Huxley currents and a stochastic spike generation process. Next, the authors constructed a simple network model using aGIF serotonin and GABA neurons, and show that heterogeneous feedforward inhibition due to heterogeneous electrophysiological properties of local GABA neurons lead to divisive inhibition of serotonin neuron firing (i.e., change in the slope of input-output function). Finally, using a ramp depolarization, the authors found that serotonin neurons encode the temporal derivative of depolarization, i.e., the slope of ramp depolarization. This property can be ascribed to the prominent spike-frequency adaptation observed in serotonin neurons.

The reviewers found the conclusions of this study very interesting and functionally relevant. The reviewers also thought that the experiments and modeling are generally sound and rigorous, and the results are presented clearly. However, the reviewers have identified various concerns and points to be clarified.

1. The slice experiments were carried out in room temperature, which distorts the kinetics of ion channels and pumps relevant for in vivo electrical activity. Ideally, some of the main conclusions should be confirmed at room temperature. In case this is difficult, please explicitly discuss this caveat and potential impacts on conclusions.

2. The series resistance up to 30 Mohm remains uncompensated and in combination with potassium currents in the range of 1 nA generates voltage errors up to 30 mV. In addition, gating properties of A-type currents will be distorted under these recording conditions. Do the main experimental results hold with a more stringent cutoff of the series resistance? Conversely, how robust are the modeling results with respect to the specific gating properties of A-type currents including the inactivation kinetics?

3. Some of the model parameters appear to be non-physiological. For example, the reversal potential of potassium is given as at least 20mV lower than that used typically in the literature. Please justify the use of -101mV rather than the range -75mV to -85mV as is more common. Please clarify whether the results depend on this very low reversal potential.

4. It is unlikely that the derivative-like computation holds in all of potential input regimes. It is important to clarify in what conditions derivative-like computation is a good approximation of the input-output relationship of serotonin neurons, and when it breaks down.

5. The derivative-like computation and its role in reinforcement learning are emphasized in the manuscript (e.g. abstract). However, as discussed above, the derivative-like computation likely occurs only in a limited input regimen, and the relevance to reinforcement learning is very speculative. The reviewers thought that these points need to be a little toned down.

*Reviewer #1 (Recommendations for the authors):*

The electrophysiological characterization of 5-HT neurons has a number of shortcomings that should be addressed by additional experiments:

1. Experiments are carried out in room temperature. This distorts the kinetics of ion channels and pumps relevant for in vivo electrical activity.

2. The series resistance up to 30Mohm remains uncompensated and in combination with potassium currents in the range of 1 nA generates voltage errors up to 30 mV. In addition, gating properties of IA-currents will be distorted under these recording conditions

3. An intracellular solution is used without calcium buffering. Given the key role of adaptation and relevant calcium-dependent conductances in 5-HT neurons this is not state-of-the-art

4. 4-AP is not a selective Kv4 (IA) channel blocker. There are more-selective, commercially available alternatives like Heteropodatoxin-2 or AmmTx3.

5. Author show that augmentation of the GIF-model with IA-currents does not specifically improve the model in comparison to a generic iGIF model (Figure 3F2). Doesn´t this call in into question the whole strategy of using experimentally validated biophysical enhancements of GIFs? Does the experimentally augmented GIF outperform the standard computational model of 5-HT neurons (see Tuckwell & Penington (2014) Computational modeling of spike generation in serotonergic neurons of the dorsal raphe nucleus. Prog Neurobiol 118:59-101).

6. Authors should experimentally study the biophysics of adaptation in 5-HT neurons as this is most related to temporal derivative computation, the key insight of this ms. For experimental conditions see 3.

*Reviewer #2 (Recommendations for the authors):*

1) Electrophysiological recordings are conducted at room temperature. It is not clear why the authors did not conduct recordings at more physiological temperature, as temperature should affect properties on many ion channels, and thus the excitability of neurons.

2) The authors may want to briefly mention the potential behavioral conditions in which endocannabinoids could be released in the dorsal raphe.

3) Dopamine neurons are also known to display low frequency firing, large afterhyperpolarizations, and A-type K currents. It would be helpful to briefly discuss whether similar computations could be applied to these neurons, especially in light of their role in reinforcement learning,

4) P. 14: Figure 1C3 is not present in Figure 1. There might be other places where figure panels are not correctly referred to.

*Reviewer #3 (Recommendations for the authors):*

The paper is very well written and clear, with a solid approach.

p.10 the reversal potential of potassium is given as at least 20mV lower than any I have come across. Please justify the use of -101mV rather than the range -75mV to -85mV as is more common.

p.11 "within 8 ms": this seems like an arbitrary time range and a rather large one to predict spike timing given typical membrane time constants and jitter. Also, it is unclear if this is related to the "1.5 ms before to 6.5 ms after" a spike where the dV/dt is constrained – and again, why this 8 ms window? And why the smaller window for SOM neurons for the fitting of dV/dt, though not the validation of spike times?

p.21-p.22 The authors state that the aGIF is more parsimonious than the iGIF but it appears that the aGIF has more parameters so the reverse would be true? Perhaps because the aGIF includes a known, well-characterized current, the number of free parameters is smaller even if the total number is greater? Or am I missing something? For sure if the current is shown to be present in the neuron then it is reasonable to use a model with that current rather than the iGIF, but "more parsimonious" would not be the reason.

The authors then on p.22 state the aGIF "best accounts" for the data, but again the iGIF appears to fit the data just as well, and no other model currents or neural were tested. I think the valid conclusion is that adding the extra current(s) improves the fit, no more. It would be good to see a criterion like AIC used to justify the improvement to fit is greater than that bound to arise with extra free parameters.

Figure 5: Given the 5-HT neuron has a sustained response that depends on step height, its output is not the differential of its input. This is an important proviso in any discussion of the properties of the circuit as a differentiator, since it is not one.

p.32, Figure 6 caption. "includes a strong subtractive component (F2)". I think "E2" may be meant as I see a shift between "Het SOM" and "Hom SOM" curves in E2 but not in F2. Also it is important to be clear when discussing the "input-output function" of a neuron whether the transient peak response is being discussed, or the sustained response, or both. Normally sustained firing rate to sustained input would be meant, so if it is the peak of the transient, that should be added in the description.

p.33 Figure 7 caption. "output is … linearly related": again, only the peak of the transient response appears to have a linear relationship. This is not the same as the neuron's output in general.

It would be nice to see that the output of the neuron responding to some range of fluctuating inputs, I(t), has higher linear correlation with dI/dt than with I or to show the neuron responds more to dI/dt than I(t) via some other statistical test. In particular, if dI/dt is more and more negative, does the rate decrease more and more (or is it only a positive slope detector)?

Figure 7F is too opaque for me to understand. It seems like two different manipulations on one plot, but then it is unclear why the different manipulations and color scales correspond to different regions of the x-y plane. I am confused, so I suggest a bit more explanation, or 2 figures if 2 manipulations are carried out.

p.39 temporal difference learning for example is not the same as producing the time-derivative of an input. It is the difference between two inputs (one being expected reward the other being actual reward) – or one can calculate it as the difference across trials from some average reward to the current reward, but that difference between the current input and trial-averaged input is over a far slower timescale than that of the peak responses to a change in input demonstrated here.

---

## [Author Response]

Essential revisions:In this study, Harkin and colleagues examined the biophysical properties of dorsal raphe (DR) serotonin neurons using a combination of whole-cell patch clamp recording in brain slices and biophysical computational modeling. The authors adopted the approach called a generalized integrate-and-fire [aGIF] model, which incorporates a relatively small number of salient biophysical properties (ionic currents) of a specific neuron, and whose parameters are optimized based on voltage dynamics obtained experimentally. First, the authors observe two prominent features in serotonin neurons. The first is a prominent spike frequency adaptation, resulting from after-hyperpolarization potentials. The second is a "kink" in the voltage trace leading up to the first spike, which can be attributed to the effect of A-type potassium currents (I-A). Using the data, the authors constructed aGIF model of serotonin neurons which contains I-A, two Hodgkin-Huxley currents and a stochastic spike generation process. Next, the authors constructed a simple network model using aGIF serotonin and GABA neurons, and show that heterogeneous feedforward inhibition due to heterogeneous electrophysiological properties of local GABA neurons lead to divisive inhibition of serotonin neuron firing (i.e., change in the slope of input-output function). Finally, using a ramp depolarization, the authors found that serotonin neurons encode the temporal derivative of depolarization, i.e., the slope of ramp depolarization. This property can be ascribed to the prominent spike-frequency adaptation observed in serotonin neurons.The reviewers found the conclusions of this study very interesting and functionally relevant. The reviewers also thought that the experiments and modeling are generally sound and rigorous, and the results are presented clearly. However, the reviewers have identified various concerns and points to be clarified.1. The slice experiments were carried out in room temperature, which distorts the kinetics of ion channels and pumps relevant for in vivo electrical activity. Ideally, some of the main conclusions should be confirmed at room temperature. In case this is difficult, please explicitly discuss this caveat and potential impacts on conclusions.

This is a valid and admittedly important point. Because this issue was presented as perhaps the chief concern related to our experimental work, we have spent considerable time and effort to experimentally address it. Experiments were initially carried out at room temperature simply because of our difficulty in obtaining high quality and stable recordings at a higher temperature. To address the possibility that some of our results could be an artifact of this experimental limitation, we optimized some slicing procedures and had a more experienced electrophysiologist in our group carry out key recordings closer to physiological temperature. We managed to carry out recordings at 29°C to 30°C measured in the recording bath. In brief, we found that the voltage-dependence and kinetics of IA were indeed slightly altered at increased temperatures, and that the very strong adaptation observed in serotonin neurons was somewhat attenuated due to a reduction in the spike-triggered threshold movement. These quantitative changes prompted us to revisit our simulations seeking to determine the output of the DRN in response to inputs of different derivatives. However, these simulations revealed that increasing temperature was not accompanied by substantive qualitative changes to our main conclusions (the simulations are different though). Since we believe other readers may share this important concern, we have added several supplementary figures to our manuscript that directly illustrate this point (*i.e.,* Figures 1S5, 2S2, 2S3, 3S3, 4S2, 5S5, and 7S1, results reported on pages 7, 15, 18, 20, and 26).

2. The series resistance up to 30 Mohm remains uncompensated and in combination with potassium currents in the range of 1 nA generates voltage errors up to 30 mV. In addition, gating properties of A-type currents will be distorted under these recording conditions. Do the main experimental results hold with a more stringent cutoff of the series resistance? Conversely, how robust are the modeling results with respect to the specific gating properties of A-type currents including the inactivation kinetics?

We thank the reviewers for explicitly noting this point and we wholeheartedly agree that the series resistance cutoff used in our initial analysis was too charitable (especially for the present case where we are trying to clamp large and fast currents). This cutoff was used for early and still expanding datasets and somehow got carried forward. Upon closer inspection prompted by this comment, we realized that the overwhelming majority of the voltage-clamp recordings used to characterize IA in 5-HT neurons had a significantly lower series resistance (14.7 +/- 6.2 MOhm, mean +/- SD). A standard series resistance cutoff of 20 MOhm ended up excluding only three of the thirteen original recordings (series resistance of the ten remaining recordings: 12.1 +/- 4.0 MOhm). We have added a note about these values on page 37 of our revised manuscript.

Do the main experimental results hold using the more stringent cutoff?

Yes. The estimates of IA peak amplitude, time to peak, and decay time constant using this more rigorous inclusion criterion are only marginally different from the original estimates (see table below). We agree with the reviewer that a more stringent cutoff would be preferable. Unfortunately, this would involve rerunning all of our simulations involving the aGIF (which is a significant portion of this study). The absence of a significant change in the kinetics strongly indicates that rerunning the simulations would not produce different results. We have made a note of the series resistance values (above) and their effects on the estimated parameters of IA ( Author response table 1) in the manuscript and kindly propose to keep the original series resistance cutoff.

**Author response table 1. sa2table1:** 

Parameter	Original estimate	Updated estimate
Amplitude (pA)	929±69	937±79
Time to peak (ms)	7.46±0.21	7.42±0.27
Time constant (ms)	42.9±2.6	41.0±3.1

Conversely, how robust are the modeling results with respect to the specific gating properties of A-type currents including the inactivation kinetics?

The modeling results do not depend on our experimental measurements of IA’s inactivation kinetics and are expected to be robust to Ra-related distortion of the equilibrium gating curves. The features of IA most important to our modeling results are that the current both activates and inactivates below spike threshold and that the inactivation time constant is not very small relative to the membrane time constant. If the ‘true’ activation and inactivation gating curves were located further to the left or rise more rapidly than the ones we report due to series resistance-related voltage errors, this current would still activate and inactivate below threshold, leaving unaltered our main conclusions. Moreover, and importantly, because the inactivation time constants of IA used in our simulations are estimated directly from the fluctuating input data as part of the GIF model fitting procedure, these values represent effective quantities that take into account the impact of IA on the subthreshold dynamics of 5-HT neurons and are not sensitive to the same sources of error as our equilibrium gating curves. (The values of the inactivation time constant estimated by the aGIF model are 53 +/- 42 ms (mean +/- SD) with an interquartile range of 27 to 77 ms (N=18), consistent with the values obtained using voltage clamp experiments). Lastly, the simulations shown in Figure 2S1 show empirically that reducing the effective inactivation time constant of IA by more than 40% does not qualitatively alter our results. Altogether, the modeling results robustly hold up to variability of gating parameters significantly beyond that expected for series resistance-mediated distortion.

3. Some of the model parameters appear to be non-physiological. For example, the reversal potential of potassium is given as at least 20mV lower than that used typically in the literature. Please justify the use of -101mV rather than the range -75mV to -85mV as is more common. Please clarify whether the results depend on this very low reversal potential.

Thank you for the opportunity to clarify this aspect of our methodology. The low potassium reversal was initially chosen to match our experimental conditions to ensure accurate model parameter estimates (we have noted this point on page 41 of our revised manuscript). Since the most important results that emerge from our simulations are related to spike timing and firing rate, which occur within the voltage range above the physiological reversal potential of potassium, raising the reversal potential of potassium would have the exact same effect as reducing the maximal conductance associated with IA (by approx. 25% to 50% depending on the reversal potential used). We have extensively explored the effects of changing this parameter in Figures 2B, 6A, 7E, and 7G, and therefore changing the reversal potential of potassium would not significantly affect our conclusions. Nonetheless, to empirically support this idea, we adjusted the potassium reversal potential to -89.1 mV (calculated using the Nernst formula and physiological potassium concentrations of [K]o = 5 mM, [K]i = 140 mM) in our aGIF models fitted to the data collected near physiological temperature. As noted in our response to point #1, the results obtained using these models are consistent with our main conclusions.

4. It is unlikely that the derivative-like computation holds in all of potential input regimes. It is important to clarify in what conditions derivative-like computation is a good approximation of the input-output relationship of serotonin neurons, and when it breaks down.

This is an important point, especially considering that 5-HT neurons can exhibit sustained firing even when the derivative of the input is zero (*e.g.,* Figure 1S1 and 5). Our results suggest that the DRN is best conceptualized as encoding *both* the intensity and temporal-derivative of its input. We have added a toy rate-based model of the DRN to capture this idea and to clarify its limitations (Figure 7S2). Reassuringly, we found that the derivative-like computation dominates DRN output precisely when the derivative of the input is large (and positive), but we have amended the results and Discussion sections to clarify when and why this computation breaks down (pages 26, 27 and 34). We have also changed our wording of this result throughout the manuscript to emphasize that the DRN encodes a *mixture* of the intensity and temporal derivative of its input (*e.g.*, abstract, page 5).

5. The derivative-like computation and its role in reinforcement learning are emphasized in the manuscript (e.g. abstract). However, as discussed above, the derivative-like computation likely occurs only in a limited input regimen, and the relevance to reinforcement learning is very speculative. The reviewers thought that these points need to be a little toned down.

Thank you for this feedback: we entirely agree that the role of this computation in RL is at this time purely speculative. After deliberation, we removed the explicit link to RL from the abstract and significantly rewrote and toned down the Discussion section on this speculative point (see pages 34, 35).

Reviewer #1 (Recommendations for the authors):The electrophysiological characterization of 5-HT neurons has a number of shortcomings that should be addressed by additional experiments:1. Experiments are carried out in room temperature. This distorts the kinetics of ion channels and pumps relevant for in vivo electrical activity.

We have experimentally addressed this point. Please see our response to *essential point #1* above.

2. The series resistance up to 30Mohm remains uncompensated and in combination with potassium currents in the range of 1 nA generates voltage errors up to 30 mV. In addition, gating properties of IA-currents will be distorted under these recording conditions

Please see our response to essential revision #2.

3. An intracellular solution is used without calcium buffering. Given the key role of adaptation and relevant calcium-dependent conductances in 5-HT neurons this is not state-of-the-art

This is an important point. 5-HT neurons are indeed endowed with several Calcium-dependent conductances that are bound to regulate their firing behaviors. As a mere example, 5-HT neurons are well known to exhibit a prominent calcium- and SK channel dependent AHP (Scuvée-Moreau et al., Br. J. Pharm, 2004; Sargin et al., *eLife* 2016). There are likely other parameters of ionic conductances expressed in 5-HT neurons that can be regulated by intracellular Calcium. As such, it is by design that we chose to omit Calcium chelators from our intracellular solution in order to have recording conditions that are as physiological as possible. In this line, the GIF formalism is designed to extract excitability parameters from spiking in such naturalistic conditions, and is thus aptly applied in conditions without Calcium buffering. Moreover, because Calcium buffering would alter the contribution of a potentially unknown set of (calcium-dependent) conductances to these spiking behaviors, these experiments would be fundamentally challenging to interpret.

4. 4-AP is not a selective Kv4 (IA) channel blocker. There are more-selective, commercially available alternatives like Heteropodatoxin-2 or AmmTx3.

We were initially primarily concerned with an algorithmic description of this conductance, and not so much with its molecular identification. That said, we followed this reviewer’s suggestion and found that Heteropodatoxin-2 and AmmTx3 block most of the transient component of the outward current readily observed in 5-HT neurons. These results are now in the revised manuscript (Figure 1S4). It is noteworthy that the delay in obtaining these particular drugs are in large part responsible for delaying the submission of this revised manuscript.

5. Author show that augmentation of the GIF-model with IA-currents does not specifically improve the model in comparison to a generic iGIF model (Figure 3F2). Doesn´t this call in into question the whole strategy of using experimentally validated biophysical enhancements of GIFs? Does the experimentally augmented GIF outperform the standard computational model of 5-HT neurons (see Tuckwell & Penington (2014) Computational modeling of spike generation in serotonergic neurons of the dorsal raphe nucleus. Prog Neurobiol 118:59-101).

Thank you for the opportunity to clarify these points. Concerning aGIF vs iGIF, although both models predict the spike trains of 5-HT neurons with comparable accuracy, we would like to emphasize that the aGIF model predicts the subthreshold voltage of these cells significantly more accurately than the iGIF model (Figure 3F1). Although accurately predicting the subthreshold voltage is certainly not always necessary (for example, the GLM and LNL-based approaches discussed on page 31, which are not viable for 5-HT neurons due to their very low firing rates, do not model subthreshold voltage at all), we contend that this represents an important feature in cases where the primary interest lies in exploring the causal relationships between mechanisms operating in subtreshold regimes and spiking behaviors. We have briefly noted this point on page 31 of our revised manuscript.

Beyond this point, the fact that aGIF and iGIF models achieve similar spike timing prediction using squarely different mechanisms has two possible interpretations: either both models capture the same variance (they explain the same phenomenon using different mechanisms) or the models complement one another (they capture different phenomena). Although we believe that the flexibility of the iGIF model makes the former more likely, if it were the latter, this would suggest that GIF models should be gradually augmented with various mechanisms, in a manner indeed reminiscent of Hodgkin-Huxely-type modelling.

There remains, however, an important drawback to biophysically-detailed Hodgkin-Huxley-style models, such as the model of Tuckwell and Penington (2014). Off-the-shelf versions of these models typically predict about half of the variance predicted by GIF models (Gerstner and Naud, Science 2009). Fitting the maximal conductances to data improves the prediction, typically reaching a performance on par with GIF models. Importantly, however, the fitting of biophysical models is exceedingly difficult and may be impossible on limited amounts of data. The fitting process for biophysically-detailed models also uses about 1000 times more computer time. As a result, even if GIF-model-based approaches appear more and more complex, they remain fundamentally tractable such that they can be calibrated on typical datasets (as we have done in this paper).

6. Authors should experimentally study the biophysics of adaptation in 5-HT neurons as this is most related to temporal derivative computation, the key insight of this ms. For experimental conditions see 3.

One of the main contributions of our GIF approach was to estimate the functional amplitude and kinetics of adaptation in 5-HT neurons directly from their spiking and subthreshold responses to in vivo-like fluctuating input (Figure 4). Somewhat surprisingly, this approach demonstrated that an increase in threshold contributed to adaptation in these neurons, in addition to the AHP that has previously been well described in these neurons. The procedure used by GIF-type models to estimate adaptation is well-established (*e.g.*, Mensi, Naud, et al., 2012, Pozzorini, Naud, et al., 2013) and does not require independent characterization of the underlying biophysical mechanisms. This is entirely different from the situation that arose with IA, where augmenting the GIF with this current required separate experiments to characterize its voltage-dependence (as briefly noted in our discussion on page 30). In an independent study, we have carried out voltage-clamp recordings that characterize IAHP in some depth (*e.g.,* modulation by apamin), but similar experiments have previously been reported and it is unclear to us how these results would be integrated into our approach (since the GIF fitting procedure would not use the experimentally-determined parameter values). Furthermore, it is not immediately apparent what type of experiments would be required to fully parameterize the dynamic change in threshold that also contributes to adaptation in these cells. Altogether, we respectfully submit that while the proposed experiments are certainly interesting, it is not clear how they would be incorporated into our model-based approach and they are therefore not required to support the main message of our work.

Reviewer #2 (Recommendations for the authors):1) Electrophysiological recordings are conducted at room temperature. It is not clear why the authors did not conduct recordings at more physiological temperature, as temperature should affect properties on many ion channels, and thus the excitability of neurons.

We have experimentally addressed this point; please see our response to *essential point #1*.

2) The authors may want to briefly mention the potential behavioral conditions in which endocannabinoids could be released in the dorsal raphe.

There is an expanding literature that is outlining with increasing mechanistic and molecular detail how endocannabinoids are release and retrogradely transmitted in the DRN (*e.g.,* Haj-Dahmane, PNAS 2018) to affect synaptic transmission (*e.g.,* Haj-Dahmane et al., JPET 2009). We have participated in this research effort by showing that glutamatergic synapses onto DRN GABAergic neurons are far more sensitive to cannabinoid-mediated inhibition than their counterpart onto 5-HT neuron (Geddes et al., PNAS 2016; this finding was one of the important motivating factor for the present study). Despite these advances, the conditions that lead to endocannabinoids release in the DRN during behavior are far less understood. For instance, while it has been shown that 5-HT neuronal activity *per se* regulates tonic release of endocannabinoid in the DRN, it is unknown whether this is also true for GABAergic neurons. Thus, beyond these broad strokes of guiding principles governing endocannabinoid release in the DRN, our knowledge is very superficial and remains mainly speculative. That said, the homogeneous rise of cannabinoid receptor activation induced by recreational cannabis use is likely to be broadly recapitulated by our modeling. We made a mention on page 34 of the revised manuscript.

3) Dopamine neurons are also known to display low frequency firing, large afterhyperpolarizations, and A-type K currents. It would be helpful to briefly discuss whether similar computations could be applied to these neurons, especially in light of their role in reinforcement learning,

We thank the reviewer for this point, this is indeed the case and increases the scope of our work. We have added a remark to that effect in the Discussion section on reinforcement learning (page 35).

4) P. 14: Figure 1C3 is not present in Figure 1. There might be other places where figure panels are not correctly referred to.

Thank you for pointing this out! We have corrected this error and reviewed the manuscript for similar issues.

Reviewer #3 (Recommendations for the authors):The paper is very well written and clear, with a solid approach.p.10 the reversal potential of potassium is given as at least 20mV lower than any I have come across. Please justify the use of -101mV rather than the range -75mV to -85mV as is more common.

Please see our response to essential revision #3.

p.11 "within 8 ms": this seems like an arbitrary time range and a rather large one to predict spike timing given typical membrane time constants and jitter. Also, it is unclear if this is related to the "1.5 ms before to 6.5 ms after" a spike where the dV/dt is constrained – and again, why this 8 ms window? And why the smaller window for SOM neurons for the fitting of dV/dt, though not the validation of spike times?

The 8ms precision used to calculate the Md* spiketrain similarity metric is admittedly somewhat arbitrary. Previous work carried out in cortical neurons used a precision of 4ms. This number was based on a systematic comparison of the spike timing precision and the intrinsic reliability (the consistency of spike timing across repetitions for a given spike timing precision) (Jolivet et al. J Neurosci Meth 2008). We have chosen Δ according to a similar assay in DRN neurons and added a note to this effect on page 42 of our revised manuscript.

The precision used in the Md* metric is unrelated to the “1.5 ms before to 6.5 ms after” window around each spike during which the voltage is ignored. Because integrate-and-fire models including GIF-class models do not explicitly model the membrane voltage during a spike, it was necessary to remove spikes from our data before attempting to fit the subthreshold components of our models. We define the timing of a spike as the instant that the membrane voltage rises past 0 mV, and creating a window of several milliseconds around this point ensures that the entire action potential is removed. The duration of this window varies slightly across cell types due to differences in action potential shape.

p.21-p.22 The authors state that the aGIF is more parsimonious than the iGIF but it appears that the aGIF has more parameters so the reverse would be true? Perhaps because the aGIF includes a known, well-characterized current, the number of free parameters is smaller even if the total number is greater? Or am I missing something? For sure if the current is shown to be present in the neuron then it is reasonable to use a model with that current rather than the iGIF, but "more parsimonious" would not be the reason.

Thank you for the opportunity for clarification. The “non-linear coupling term” that differentiates the GIF and iGIF models mentioned in our results text actually consists of a non-parametric function of past voltage. The non-parametric definition of this term introduces roughly ten additional free parameters compared with the GIF and aGIF models (our aGIF model has only three additional free parameters compared with the GIF model), and, as with non-parametric models in general, the fitted values of these parameters can be difficult to interpret. We believe that this makes the aGIF model more parsimonious and have added clarification to the methods and Results sections to emphasize this point (pages 39 and 14 in the revised manuscript).

The authors then on p.22 state the aGIF "best accounts" for the data, but again the iGIF appears to fit the data just as well, and no other model currents or neural were tested. I think the valid conclusion is that adding the extra current(s) improves the fit, no more. It would be good to see a criterion like AIC used to justify the improvement to fit is greater than that bound to arise with extra free parameters.

Please see point no. 5 from reviewer 1 for changes related to this point. In writing the manuscript, we were conscious of the equal predictive performance of the aGIF and iGIF (specifically with respect to spike-timing; the aGIF better predicts the subthreshold dynamics of 5-HT neurons), which led us to attempt to make the more limited claim that the aGIF model best accounts for the *biophysical mechanisms* responsible for shaping the spiking behaviour of 5-HT neurons rather than the data (see page 15). Regarding the AIC, however, we respectfully disagree as this methodology was invented for cases where data that was not used for model fitting is not available (i.e., test or validation data). Our data collection approach was designed to collect a separate validation dataset and obviate the need for statistical tools such as AIC (Figure 3C and D). Cross-validation remains a more accurate approach to model selection that takes into account overfitting.

Figure 5: Given the 5-HT neuron has a sustained response that depends on step height, its output is not the differential of its input. This is an important proviso in any discussion of the properties of the circuit as a differentiator, since it is not one.

We believe that DRN is indeed best conceptualized as encoding a mixture of the intensity and temporal derivative of its input. We have added a toy model to formalize this intuition (Figure 7S2) and have adjusted the manuscript text accordingly. (See our responses to other points.)

p.32, Figure 6 caption. "includes a strong subtractive component (F2)". I think "E2" may be meant as I see a shift between "Het SOM" and "Hom SOM" curves in E2 but not in F2. Also it is important to be clear when discussing the "input-output function" of a neuron whether the transient peak response is being discussed, or the sustained response, or both. Normally sustained firing rate to sustained input would be meant, so if it is the peak of the transient, that should be added in the description.

Thank you for pointing this out, we have corrected the issue.

p.33 Figure 7 caption. "output is … linearly related": again, only the peak of the transient response appears to have a linear relationship. This is not the same as the neuron's output in general.

Agreed. We have adjusted the wording of the caption to specify the peak firing rate and added Figure 7S2 to support our broader point.

It would be nice to see that the output of the neuron responding to some range of fluctuating inputs, I(t), has higher linear correlation with dI/dt than with I or to show the neuron responds more to dI/dt than I(t) via some other statistical test. In particular, if dI/dt is more and more negative, does the rate decrease more and more (or is it only a positive slope detector)?

Thank you for the suggestion. We had initially considered taking a similar approach, but were concerned that quantifying the correlation between the output and dI/dt across multiple timescales and levels of background input would introduce sufficient analytical complexity to alienate part of our target readership (and perhaps ourselves). To add clearer support for our derivative-encoding argument, we have fitted a toy rate model of DRN output as a function of the input and its derivative to the data presented in Figure 7 and compared it with a restricted model that does not take the derivative of the input into account (Figure 7S2). We hope that the stark differences between the predictions made by the two toy models will be an adequate compromise to address this point.

Figure 7F is too opaque for me to understand. It seems like two different manipulations on one plot, but then it is unclear why the different manipulations and color scales correspond to different regions of the x-y plane. I am confused, so I suggest a bit more explanation, or 2 figures if 2 manipulations are carried out.

The goal of this figure was to show that two of the biophysical features of 5-HT neurons contribute to shaping the input-output function of the DRN under distinct input regimes. We have updated the caption to emphasize this point and hope that it is now clearer.

p.39 temporal difference learning for example is not the same as producing the time-derivative of an input. It is the difference between two inputs (one being expected reward the other being actual reward) – or one can calculate it as the difference across trials from some average reward to the current reward, but that difference between the current input and trial-averaged input is over a far slower timescale than that of the peak responses to a change in input demonstrated here.

Thank you for raising this point. We have made changes to that effect, see *essential point 5*.